# RAL GTPases mediate EGFR-driven intestinal stem cell proliferation and tumourigenesis

**Máté Nászai[1,2], Karen Bellec[1,2], Yachuan Yu[1,2,3], Alvaro Román-Fernández[2,3], Emma Sandilands[2,3], Joel Johansson[3], Andrew D Campbell[3], Jim C Norman[2,3], Owen J Sansom[2,3], David M Bryant[2,3], Julia B Cordero[1,2,3]***

[1]Wolfson Wohl Cancer Research Centre, Glasgow, United Kingdom; [2]Institute of Cancer Sciences, University of Glasgow, Glasgow, United Kingdom; [3]Cancer Research UK Beatson Institute, Glasgow, United Kingdom

**Abstract** RAS-like (RAL) GTPases function in Wnt signalling-dependent intestinal stem cell proliferation and regeneration. Whether RAL proteins work as canonical RAS effectors in the intestine and the mechanisms of how they contribute to tumourigenesis remain unclear. Here, we show that RAL GTPases are necessary and sufficient to activate EGFR/MAPK signalling in the intestine, via induction of EGFR internalisation. Knocking down *Drosophila RalA* from intestinal stem and progenitor cells leads to increased levels of plasma membrane-associated EGFR and decreased MAPK pathway activation. Importantly, in addition to influencing stem cell proliferation during damage-induced intestinal regeneration, this role of RAL GTPases impacts on EGFR-dependent tumourigenic growth in the intestine and in human mammary epithelium. However, the effect of oncogenic RAS in the intestine is independent from RAL function. Altogether, our results reveal previously unrecognised cellular and molecular contexts where RAL GTPases become essential mediators of adult tissue homeostasis and malignant transformation.

*For correspondence:
julia.cordero@glasgow.ac.uk

## Introduction

The precise spatial and temporal regulation of signalling pathway activity is essential for organ development and adult tissue homeostasis. The latter is particularly important in stem cell maintained self-renewing epithelia, such as that of the gastrointestinal tract (*Richardson et al., 2014*), where cell loss needs to be counteracted by stem cell proliferation and differentiation while limiting the potential for unwanted overgrowth (*Radtke and Clevers, 2005*). Progressive loss of control over proliferative pathways either through loss of tumour suppressor genes or the activation of oncogenes is associated with tumour development and progression (*Hanahan and Weinberg, 2011*).

Regulation of intestinal homeostasis involves the coordinated action of multiple evolutionarily conserved signalling pathways, which relay environmental and niche-derived signals to stem cells to ultimately determine their activity (*Gehart and Clevers, 2019*; *Nászai et al., 2015*; *Scoville et al., 2008*). Increasing understanding of how these pathways are regulated not only provides insight into basic stem cell biology, but also sheds light onto pathological conditions often associated with uncontrolled stem cell proliferation, such as cancer (*Biteau et al., 2011*; *Sell, 2010*).

Epidermal growth factor receptor (EGFR, also known as ErbB1 or HER1) is a member of the ErbB family of growth factor receptors, which play essential roles in regulating cell proliferation, differentiation, and survival (*Citri and Yarden, 2006*; *Wee and Wang, 2017*). In the mammalian intestinal epithelium, EGFR is highly expressed in intestinal stem cells (ISCs) and transit-amplifying cells (*Yang et al., 2017*). EGFR ligands, such as EGF, are released by Paneth cells and the mesenchyme and are required for the maintenance and proliferation of ISCs (*Dvorák et al., 1994*; *Jardé et al.,*

*2020*; *Poulsen et al., 1986*). Ectopic activation of EGFR signalling in the intestine by luminal application or genetic overexpression of pathway ligands (*Bongers et al., 2012*; *Kitchen et al., 2005*; *Marchbank et al., 1995*), or deletion of the negative regulator leucine-rich repeats and immunoglobulin-like domains protein 1 (Lrig1) (*Powell et al., 2012*; *Wong et al., 2012*), leads to elevated ISC proliferation. On the other hand, loss of EGFR signalling induces quiescence of Lgr5 + ISCs in vitro (*Basak et al., 2017*).

Gene amplification and activating point mutations of EGFR are highly prevalent in cancer (*Santarius et al., 2010*; *Yarden and Pines, 2012*). Ectopic EGFR/Ras/MAPK signalling is thought to be an early step in colorectal cancer (CRC) development (*Calcagno et al., 2008*). Hyperactivation of the pathway accelerates intestinal tumourigenesis driven by *Adenomatous polyposis coli* loss (*Apc^min/+* mice) (*Luo et al., 2009*), while a genetic background of partial loss-of-function of EGFR (*Roberts et al., 2002*) or small-molecule inhibitor treatment reduces cancer incidence (*Roberts et al., 2002*; *Torrance et al., 2000*).

The *Drosophila* intestinal epithelium shares remarkable homology with its mammalian counterpart. The tissue is maintained by ISCs that replenish the epithelium through progenitor cells called enteroblasts (EBs), which differentiate into either secretory enteroendocrine (EE) cells or absorptive enterocytes (ECs) (*Micchelli and Perrimon, 2006*; *Ohlstein and Spradling, 2006*). Importantly, signalling pathways governing intestinal proliferation and differentiation are highly conserved between fruit flies and mammals (*Nászai et al., 2015*; *Miguel-Aliaga et al., 2018*). Activation of EGFR/Ras/MAPK within ISCs by niche-derived EGF-like ligands is essential to sustain homeostatic and regenerative proliferation of the adult fly midgut, while constitutive pathway activation in ISCs is sufficient to drive intestinal hyperplasia (*Biteau and Jasper, 2011*; *Buchon et al., 2010*; *Jiang et al., 2011*; *Xu et al., 2011*).

Regulation of EGFR signalling activity is highly dependent on various modes of receptor trafficking throughout the endocytic pathway. Indeed, abnormal trafficking of receptor tyrosine kinases is linked to cancer (*Lanzetti and Di Fiore, 2017*; *Mosesson et al., 2008*). Following internalisation through Clathrin-mediated (CME) or Clathrin-independent endocytosis (CIE) (*Sorkin and Goh, 2009*), EGF ligand/receptor complexes can either be targeted for recycling into the plasma membrane (PM) or ubiquitinated and targeted to late endosomes for lysosomal degradation (*Sigismund et al., 2008*; *Sigismund et al., 2013*). Most recently, autophagy has emerged as an important mechanism implicated in the termination of EGFR/MAPK signalling in the intestine (*Zhang et al., 2019*). While endocytosis is classically considered as a process to terminate pathway activity (*Tomas et al., 2014*), significant evidence suggests that receptors retain their ability to relay their signal even after internalisation, hence signalling is not limited to the PM (*Sadowski et al., 2009*). The relative contribution of PM versus intracellular EGFR to downstream signalling in vivo remains unclear (*Sousa et al., 2012*; *Teis et al., 2006*).

RAL small GTPases are best recognised for their role as effectors of Ras signalling, which has attracted basic and translational research into their potential in cancer development and progression (*Moghadam et al., 2017*). Mammalian RAL GTPases, RALA and RALB, have well-characterised roles in membrane trafficking through their involvement in the exocyst complex (*Bodemann and White, 2008*; *Chen et al., 2007*; *Chien et al., 2006*) and in the regulation of Clathrin (*Jullien-Flores et al., 2000*) and caveolar-dependent endocytosis (*Jiang et al., 2016*). RAL signalling is potentiated by RALGEFs and negatively regulated by RALGAPs (*Neel et al., 2011*). RALGEF, such as RALGDS, can be activated upon association with oncogenic RAS (*Koyama and Kikuchi, 2001*) and mediate Ras-driven skin tumourigenesis (*González-García et al., 2005*).

We recently identified a novel role of RAL GTPases in the regulation of Wnt signalling activity in ISCs through the regulation of Wnt receptor trafficking into intracellular compartments (*Johansson et al., 2019*). The relevance of RAL GTPases in intestinal tumourigenesis remained unaddressed as their function in the intestine became redundant upon loss of *Apc*, a key driver of CRC (*Johansson et al., 2019*). Furthermore, whether RAL proteins (RALs) can impact intestinal biology beyond Wnt signalling and through their classical role as Ras effectors is unclear.

Here, using the *Drosophila* intestine and human lung and breast cancer cell lines we uncover an important role of RAL GTPases activating EGFR/MAPK signalling-driven cell proliferation through induction of EGFR internalisation. Our results show that, while RAL inhibition is an efficient means of attenuating intestinal hyperplasia caused by constitutively active forms of *EGFR*, the effect of oncogenic *Ras* in the intestine is insensitive to attenuation of RAL function. Our findings support a

positive role of receptor tyrosine kinase internalisation in signalling activation in vivo and identify physiological and pathological settings highly sensitive to the presence of RAL proteins, which may provide ideal platforms for the development of therapeutic approaches geared towards the modulation of RAL function.

## Results

### RAL GTPases are necessary for EGFR/MAPK signalling activation following damage to the intestinal epithelium

We have previously demonstrated that *RalA,* the single *Ral* gene in *Drosophila*, is required for Wnt signalling activation in the developing *Drosophila* wing and adult midgut (*Johansson et al., 2019*). A canonical role of *RalA* as RAS effector remained unaddressed.

EGFR/Ras signalling is an important determinant of wing tissue patterning (*Wang et al., 2000*; *Zecca and Struhl, 2002*) and ISC proliferation in the adult *Drosophila* midgut (*Biteau and Jasper, 2011*; *Buchon et al., 2010*; *Jiang et al., 2011*; *Xu et al., 2011*). We observed that adult wings resulting from RNAi-driven knockdown of *RalA* using the *engrailed-gal4* driver (*en>RalA* RNAi) showed a more severely dysmorphic phenotype than that caused by *wingless* knockdown (*en>wg* RNAi) or *EGFR* knockdown (*en>EGFR* RNAi) only (*Figure 1A, B*). Instead, adult wings from *en>RalA* RNAi animals displayed a dysmorphic phenotype more similar to that resulting from combined knockdown of both *wg* and *EGFR* downregulation (*en>wg RNAi+EGFR* RNAi) (*Figure 1A, B*). These results led us to hypothesise that *RalA* may regulate pathways other than Wnt signalling, including EGFR/Ras signalling. To address this, we turned to the adult *Drosophila* midgut, a robust paradigm for the study of signal transduction in adult tissue homeostasis, where *RalA* plays a pivotal role (*Johansson et al., 2019*).

*RalA* is required within ISCs to induce adult midgut regeneration following damage by oral infection with *Erwinia carotovora carotovora 15 (Ecc15)* (*Johansson et al., 2019*). To achieve a global view of intestinal pathways affected by RalA, we performed a transcriptomic analysis by RNAseq of whole midguts from vehicle-treated (Mock) or damaged (*Ecc15* fed) control animals or following adult-restricted *RalA* knockdown in intestinal stem and progenitor cells using the e*scargot-gal4* driver (*ISC/EB>*) (*Micchelli and Perrimon, 2006*). Consistent with its effect on ISC proliferation (*Johansson et al., 2019*), *RalA* knockdown significantly impaired damage-induced upregulation of cell cycle genes in the midgut (*Figure 1C*). Additionally, levels of genes associated with the EGFR/MAPK pathway, such as *argos* (*aos*), *rhomboid* (*rho*), *Sox21a,* and *string* (*stg*), appeared increased following *Ecc15* infection in control midguts in a *RalA*-dependent manner (*Figure 1C*). RT-qPCR confirmed RNAseq results on *rho*, a well-characterised activator of EGFR/MAPK signalling in ISCs (*Liang et al., 2017*; *Ngo et al., 2020*), and two downstream targets of the pathway required for ISC proliferation, *Sox21a* and *stg* (*Jin et al., 2015*; *Meng and Biteau, 2015*; *Figure 1D*). Furthermore, immunofluorescence staining for the transcription factor Sox21a (*Meng and Biteau, 2015*) and the activated form of the MAPK, phosphorylated ERK (pERK), in control animals and following *RalA* knockdown from ISCs/EBs confirmed the need for *RalA* for upregulation of MAPK signalling and downstream targets following damage to the midgut (*Figure 1E–H* and *Figure 1—figure supplement 1A–D*). Together, these results suggest that *RalA* is necessary for damage-induced EGFR/MAPK signalling activation in the *Drosophila* adult midgut.

Previously, we showed that the role of RAL proteins in Wnt signalling activation and intestinal regeneration is conserved between *Drosophila* and mice (*Johansson et al., 2019*). The mouse intestine has a robust capacity to regenerate following damage by gamma irradiation, as demonstrated by an increase in the number of regenerating crypts 72 hr following irradiation (*Cordero et al., 2014*; *Johansson et al., 2019*). We next assessed whether MAPK activation in the regenerating mouse intestine required RAL GTPases. Single conditional knockout of either *Rala* (*Rala*<sup>fl/fl</sup>) or *Ralb* (*Ralb*<sup>fl/fl</sup>) in the murine intestinal epithelium using the *Villin-CreER* driver impaired ERK activation in regenerating intestines when compared to control (*VillinCre*<sup>ER</sup>) (*Figure 1I, J* and *Figure 1—figure supplement 1E*). Therefore, RAL GTPases' requirement for EGFR/MAPK pathway activation in the intestinal epithelia is evolutionarily conserved between fruit flies and mammals.

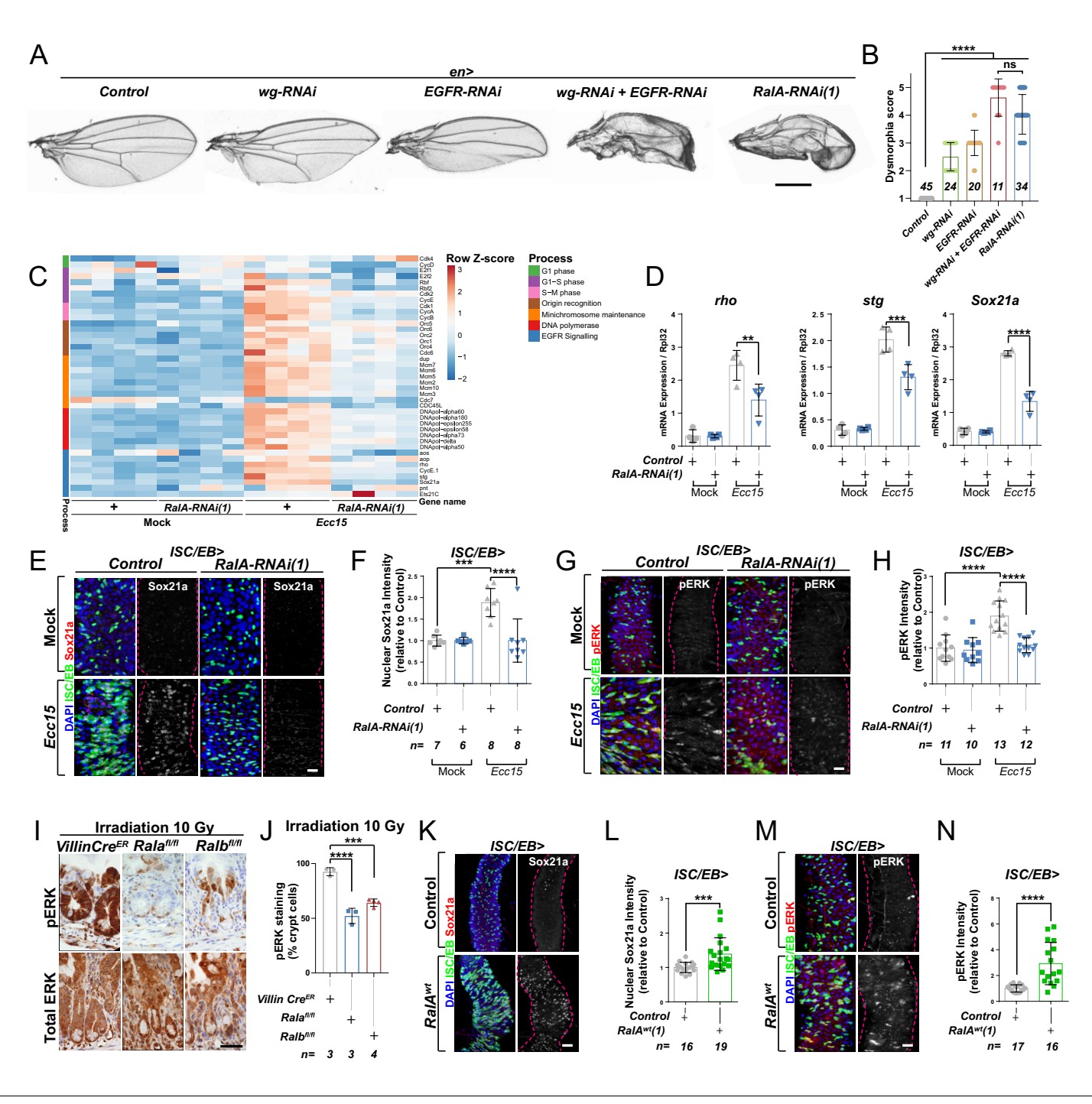

**Figure 1.** Ral GTPases are necessary and sufficient to induce EGFR/MAPK signalling in intestinal stem cells (ISCs). (**A**) Adult *Drosophila* wings from control animals and with posterior compartment knockdown of *wg* (*wg-RNAi*), *Egfr* (*Egfr-RNAi*), or *RalA* using one of two previously validated RNAi lines (*RalA-RNAi(1)*) or combined *wg* and *Egfr* knockdown (*wg-RNAi+Egfr* RNAi). Scale bar = 500 µm. (**B**) Blind scoring of wing dysmorphia on a scale of 1–5. Numbers inside bars represent the total number of wings scored. Kruskal–Wallis test followed by Dunn's multiple comparisons test. (**C**) Heat map from transcriptomic analysis of adult whole midguts from mock-treated and *Ecc15*-infected control animals (+) or following adult-restricted knockdown of *RalA* (*RalA-RNAi(1)*) using the *escargot-gal4, UAS-gfp* driver (ISC/EB>). RNA was extracted from >25 whole midguts per replicate, and four biological replicates per genotype/per condition were processed for sequencing. (**D**) RT-qPCR confirmation of genes associated with EGFR/MAPK signalling in whole midguts from genotypes and conditions as in (**C**) expressed relative to *rpl32* levels. n (number of biological replicates) = 4, each dot represents an independent RNA sample from >25 midguts per sample. Two-way ANOVA followed by Sidak's multiple comparisons test. (**E**) Representative

*Figure 1 continued on next page*

Figure 1 continued

confocal images of Sox21a immunofluorescence staining (red/grey) of adult posterior midguts from Mock-treated or *Ecc15*-infected wild-type control animals or following knockdown of *RalA* (*RalA-RNAi(1)*) in stem/progenitor cells using *escargot-gal4, UAS-gfp* (ISC/EB>; green). (F) Quantification of average Sox21a staining intensity within the nuclear compartment (DAPI positive) in midguts as in (E). Two-way ANOVA followed by Sidak's multiple comparisons test; n = number of z-stack confocal images quantified, each from an independent posterior midgut. (G) Representative confocal images of pERK immunofluorescence staining (red/grey) of adult posterior midguts from Mock-treated or *Ecc15*-infected control animals or following knockdown of *RalA* (*RalA-RNAi(1)*) within stem/progenitor cells (ISC/EB>; green). (H) Quantification of average pERK staining intensity within the ISC/EB compartment (GFP positive) of midguts as in (G). Two-way ANOVA followed by Sidak's multiple comparisons test; n = number of z-stack confocal images quantified, each from an independent posterior midgut. (I) Immunohistochemistry images of total (bottom panels) and pERK (top panels) in small intestinal regenerating crypts 3 days after whole-body irradiation of control mice (left panels) or mice following conditional intestinal epithelial knockout of *Rala* or *Ralb*. Scale bar = 50 µm. (J) Quantification of the percentage of cells with pERK staining in regenerating small intestinal crypts as in (I). n = number of mice, with >12 crypts quantified per animal, each dot represents the average percentage from a given mouse. One-way ANOVA followed by Tukey's multiple comparisons test. (K) Representative confocal images of Sox21a immunofluorescence staining (red/grey) of adult posterior midguts from control animals or animals overexpressing wild-type *Rala* within stem/progenitor cells (ISC/EB>; green). Scale bar = 50 µm. (L) Quantification of average Sox21a staining intensity within the nuclear compartment (DAPI positive; blue) of midguts as in (K). Student's t-test; n = number of z-stack confocal images quantified, each from an independent posterior midgut. (M) Representative confocal images of pERK immunofluorescence staining (red/grey) in control animals or animals overexpressing wild-type *Rala* within stem/progenitor cells (ISC/EB>; green). (N) Quantification of average pERK staining intensity within the ISC/EB compartment (GFP positive) of midguts as in (M). Student's t-test; n = number of z-stack confocal images quantified, each from an independent posterior midgut. Where indicated: *p<0.05, **p<0.01, ***p<0.001, ****p<0.0001, ns: not significant. All error bars represent SD. Scale bars = 20 µm, unless otherwise stated.

The online version of this article includes the following source data and figure supplement(s) for figure 1:

**Source data 1.** Ral GTPases are necessary and sufficient to induce EGFR/MAPK signalling in intestinal stem cells.
**Figure supplement 1.** Ral GTPases are necessary and sufficient to induce EGFR/MAPK signalling in intestinal stem cells (ISCs).
**Figure supplement 1—source data 1.** Ral GTPases are necessary and sufficient to induce EGFR/MAPK signalling in intestinal stem cells.

## RAL GTPases are sufficient for EGFR/MAPK signalling activation in the *Drosophila* midgut

Ectopic expression of wild-type *RalA* in ISC/EB is sufficient to induce Wnt pathway activation and intestinal proliferation in the *Drosophila* midgut (*Johansson et al., 2019*). To determine whether RalA is also sufficient to induce EGFR/MAPK signalling, we assessed Sox21a (*Figure 1K, L*), pERK (*Figure 1M, N*), and total ERK (*Figure 1—figure supplement 1F, G*) levels by immunostaining following *RalA* overexpression in midgut ISCs/EBs. While levels of Sox21a and pERK were increased in *RalA* overexpressing midguts compared to wild-type control ones (*Figure 1K–N*), total levels of ERK in the midgut remained unchanged across genotypes (*Figure 1—figure supplement 1F, G*). Immunostaining results for ERK and pERK were confirmed by western blot (*Figure 1—figure supplement 1H*) and are consistent with ERK activation and not total protein levels being increased upon midgut injury (*Figure 1—figure supplement 1H–J*). Altogether, our data suggest that RAL GTPases are necessary and sufficient for EGFR/MAPK pathway activation within the intestinal epithelium.

## RalA activation is necessary for ISC proliferation in *Drosophila*

Small GTPases cycle between two alternative conformations: inactive (GDP-bound) and active (GTP-bound). The balance between these states is determined by the activity of guanine nucleotide exchange factors (GEF) and GTPase activating proteins (GAP), which activate and inactivate GTPases, respectively (*Neel et al., 2011*). There are seven Ral GEFs in the human genome, *RALGDS*, *RALGPS1-2*, and *RGL1-4*, which are often found misregulated in cancer (*González-García et al., 2005*; *Koyama and Kikuchi, 2001*; *Rodriguez-Viciana and McCormick, 2005*) and are considered emerging therapeutic targets (*Neel et al., 2011*; *Vigil et al., 2010*). However, the in vivo role of RAL GEFs in the intestine remains unknown. Several Ral GEFs are conserved in *Drosophila* (*Gentry et al., 2014*): *Rgl*, *GEFmeso* and *CG5522* (*RalGPS*). *Rgl* is a close orthologue of mammalian *RGL* (*Mirey et al., 2003*), *GEFmeso* was identified in a yeast two hybrid screen using active RalA as bait (*Blanke and Jäckle, 2006*), while *CG5522* was identified based on its close homology to mammalian *RalGPS1* (*Hu et al., 2011*).

We next tested the functional role of each of these Ral GEFs in the fly midgut though RNAi-driven targeted knockdown and assessment of their impact on intestinal regeneration following oral infection with *Ecc15* (*Basset et al., 2000*). The regenerative capacity of the adult posterior midgut (R4-

R5) was quantified as per the number of proliferating ISCs, identified by staining with phosphory-lated histone H3 antibody (pH3). As expected, *Ecc15* infection induced significant increase in ISC proliferation relative to mock-treated control animals (*Figure 2A–D*). Knocking down either of the three Ral GEFs of interest significantly impaired regenerative ISC proliferation in the midgut (*Figure 2A–D*) to levels comparable to those observed upon *RalA* knockdown (*Johansson et al., 2019*). Furthermore, Ral GEF knockdown led to a significant reduction in MAPK activation in the midgut following damage (*Figure 2E, F*). These results provide evidence highlighting the impor-tance of maintaining the active status of RalA for robust stem cell proliferation and MAPK activation in the intestine.

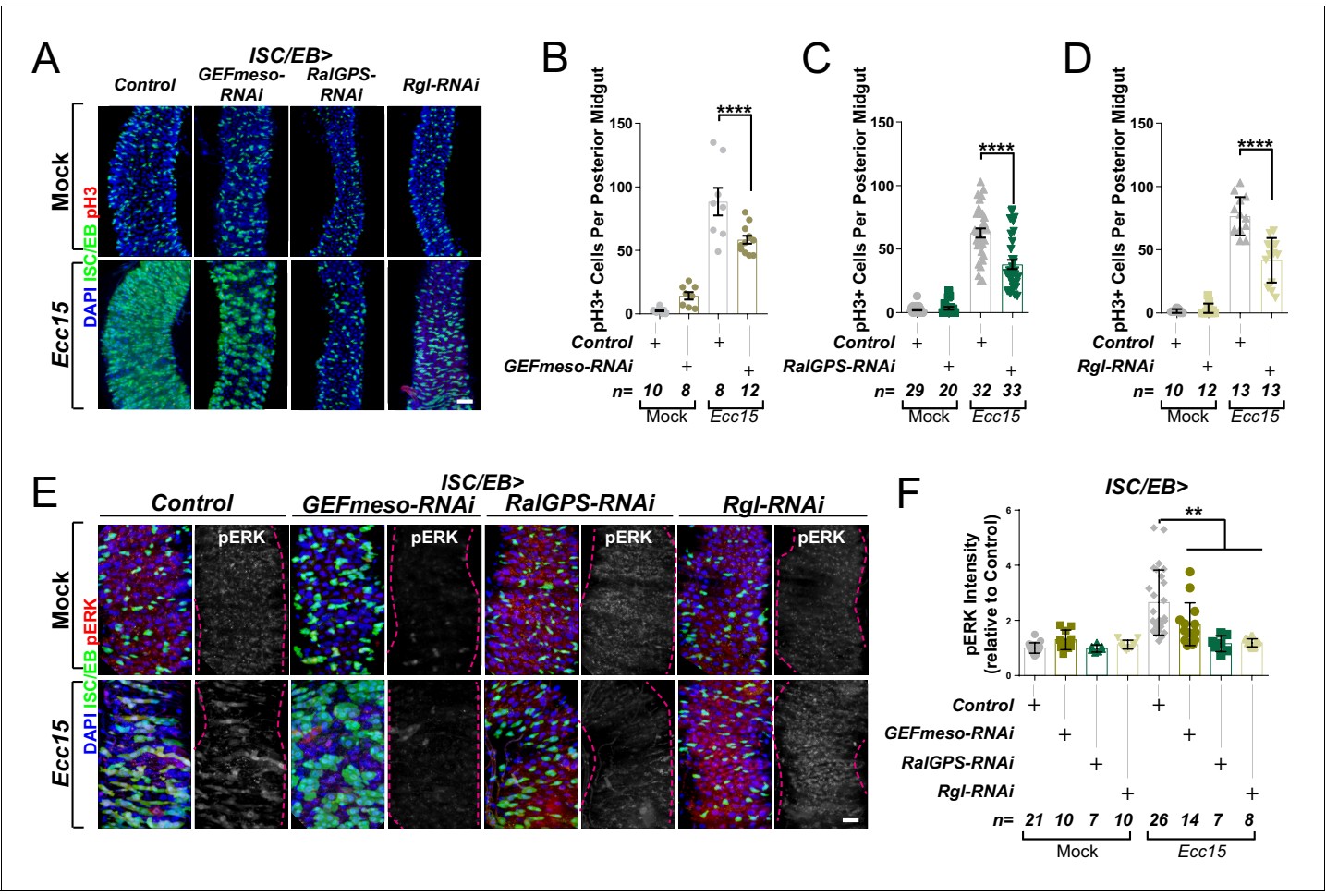

**Figure 2.** Ral GTPase activation is necessary for EGFR/MAPK signalling in regenerating intestinal stem cells/enteroblasts (ISCs/EBs). (A) Representative confocal images of pH3 staining (red) within the ISC/EB compartment (green) in mock-treated or regenerating posterior midguts. Scale bar = 50 μm. (B) Quantification of pH3-positive nuclei in control or *GEFmeso-RNAi* posterior midguts as in (A). Two-way ANOVA followed by Sidak's multiple comparisons test. n = number of midguts. (C) Quantification of pH3-positive nuclei in control or *RalGPS-RNAi* posterior midguts as in (A). Two-way ANOVA followed by Sidak's multiple comparisons test. n = number of midguts. (D) Quantification of pH3-positive nuclei in control or *Rgl-RNAi* posterior midguts as in (A). Two-way ANOVA followed by Sidak's multiple comparisons test. n = number of midguts. (E) Representative confocal images of pERK staining (red/grey) in mock-treated or regenerating control animals or animals with knockdown of *GEFmeso, RalGPS,* or *Rgl* within the ISC/EB compartment (green). Scale bar = 20 μm. (F) Quantification of average pERK staining intensity within the ISC/EB compartment (GFP positive) as in (E). Two-way ANOVA followed by Sidak's multiple comparisons test; n = number of z-stack confocal images quantified, each from an independent posterior midgut. Where indicated: *p<0.05, **p<0.01, ***p<0.001, ****p<0.0001, ns: not significant. All error bars represent SD. Scale bars = 20 μm, unless otherwise stated.

The online version of this article includes the following source data for figure 2:

**Source data 1.** Ral GTPase activation is necessary for EGFR/MAPK signalling in regenerating ISCs/EBs.

## RalA regulates EGFR- but not oncogenic Ras-driven hyperplasia in the intestine

During our initial assessment of genetic interactions between EGFR signalling and RalA in adult wings, we observed that constitutive overexpression of *EGFR* under *engrailed-gal4* (*en>EGFR^wt^*) caused severe organismal lethality, which was greatly suppressed by concomitant knockdown of *RalA* (*Figure 3—figure supplement 1A*). Wing vein patterning defects observed in rare *en>EGFR^wt^* adult escapers was also suppressed by *RalA* knockdown (*Figure 3—figure supplement 1B*). These results reinforced the importance of *RalA* as a broad mediator of EGFR signalling.

EGFR is overexpressed in ~20% of breast and ~80% of CRCs (*Rimawi et al., 2010*; *Spano et al., 2005*), and activating mutations of Ras are one of the most common cancer-associated genetic alterations (*Prior et al., 2012*). Activation of the EGFR/MAPK pathway in the adult *Drosophila* midgut by ISC/EB-specific overexpression of wild-type EGFR (*EGFR^WT^*) or constitutively active Ras (*Ras^V12^*) was sufficient to induce intestinal hyperproliferation (*Figure 3A, B*; *Jiang et al., 2011*; *Zhang et al., 2019*). Downregulation of *RalA* suppressed *EGFR^WT^*- but not *Ras^V12^*-driven ISC hyperproliferation (*Figure 3A, B* and *Figure 3—figure supplement 1C, D*). Consistently, *RalA* knockdown impaired activation of ERK following *EGFR^wt^*, but not *Ras^V12^* overexpression (*Figure 3C, D* and *Figure 3—figure supplement 1E, F*).

## RalA potentiates EGFR signalling activity downstream of ligand binding

Increasing the pool of receptors available for ligand binding, such as through recycling of intracellular receptor towards the PM or inhibition of receptor degradation, favours activation of receptor tyrosine kinase signalling, including EGFR (*von Zastrow and Sorkin, 2007*; *Zhang et al., 2019*). Therefore, one possible mechanism by which RAL proteins may potentiate EGFR signalling in the intestine is by facilitating ligand/receptor interactions. In that case, ligand-independent, constitutively active forms of EGFR, which are linked to cancer (*Endres et al., 2014*), should be insensitive to RAL deficiency. To test this prediction, we co-expressed *RalA-RNAi* with two active mutant forms of EGFR – *EGFR^λtop^* and *EGFR^A887T^* – in *Drosophila* intestinal stem and progenitor cells (*Figure 3E, F*). *EGFR^λtop^* includes an extracellular dimerisation domain that causes receptor activation even in the absence of ligand (*Queenan et al., 1997*), and *EGFR^A887T^* contains an activating point mutation in the receptor kinase domain (*Lesokhin et al., 1999*). Importantly, overexpression of *EGFR^λtop^* or *EGFR^A887T^* led to ISC hyperproliferation levels comparable to those observed following *Ras^V12^* overexpression (*Figure 3E, F* compare with *Figure 3A, B* and *Figure 3—figure supplement 1C, D*). However, unlike in the case of *Ras^V12^*, knocking down *RalA* significantly impaired *EGFR^λtop^*- or *EGFR^A887T^*-driven ISC proliferation (*Figure 3E, F*). Consistently, *EGFR^λtop^*- or *EGFR^A887T^*-dependent ERK activation was also suppressed by *RalA-RNAi* (*Figure 3G, H*). These results suggest that *RalA* influences EGFR signalling activity downstream of ligand/receptor binding.

## RAL GTPases are required for EGFR internalisation

RAL GTPases are key mediators of Ras-regulated membrane trafficking (*Bodemann and White, 2008*; *Chen et al., 2007*; *Chien et al., 2006*; *Jiang et al., 2016*; *Jullien-Flores et al., 2000*). We next asked whether, as in the case of the Wnt receptor Frizzled (*Johansson et al., 2019*), RAL GTPases may induce EGFR/MAPK signalling through regulation of EGFR cellular localisation in the intestine. We used a well-established immunostaining approach (*Cordero et al., 2014*; *Kim-Yip and Nystul, 2018*; *Zhang et al., 2019*) and a custom-developed macro to visualise EGFR cellular localisation in the adult *Drosophila* midgut (*Figure 4—figure supplement 1*). Firstly, we assessed EGFR localisation in control adult *Drosophila* midguts or following genetic manipulation of *RalA* expression. Knocking down *RalA* in ISCs/EBs led to significantly increased levels of PM-associated EGFR wild-type (*Figure 4A, B*) and A887T mutant (*Figure 4C, D*). Conversely, overexpression of wild-type RalA decreased membrane localisation of EGFR (*Figure 4E, F*). We were unable to assess the impact of knocking down *RalA* on *EGFR^λtop^* localisation as our antibody, designed to bind the extracellular domain of EGFR, failed to recognise this mutant version of the receptor. Consistent with the role of RAL GTPases as effectors of Ras, knocking down endogenous *Ras* from ISCs/EBs caused a similar effect on EGFR localisation than that observed upon *RalA* downregulation (*Figure 4—figure supplement 2*). Altogether, these results strongly suggest that activation of RalA induces EGFR/MAPK signalling in the intestine by increasing the intracellular pool of EGFR.

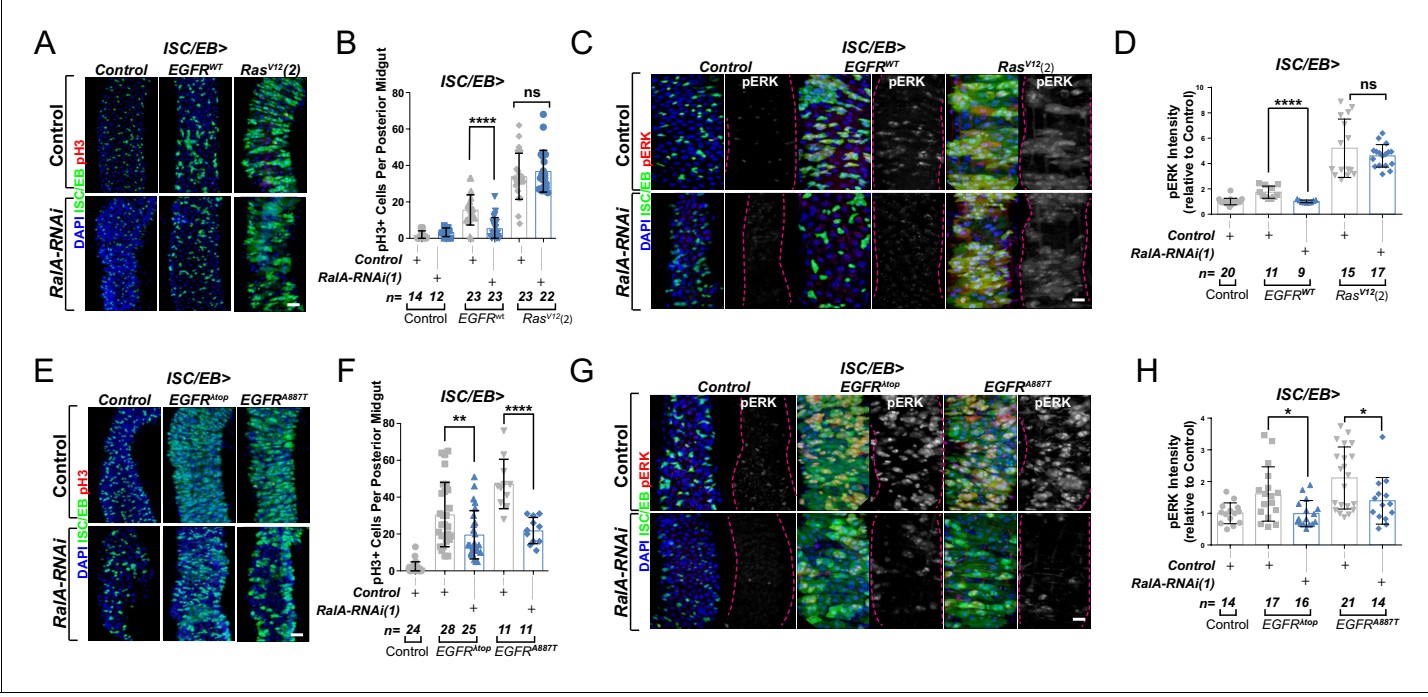

**Figure 3.** Ral GTPases are required for EGFR/MAPK signalling upstream of Ras. (**A**) Representative confocal images of pH3 staining (red) within the intestinal stem cell/enteroblast (ISC/EB) compartment (green) of control animals or animals overexpressing wild-type *Egfr* (*EGFR^WT^*) or one of two constitutive *Ras* constructs used in this paper (*Ras^V12^(2)*) with or without *RalA* knockdown within stem/progenitor cells (ISC/EB>; green). Scale bar = 50 µm. (**B**) Quantification of pH3-positive nuclei in posterior midguts as in (**A**). Two-way ANOVA followed by Sidak's multiple comparisons test. n = number of midguts. (**C**) Representative confocal images of pERK staining (red/grey) of control animals or animals overexpressing wild-type *Egfr* (*EGFR^WT^*) or one of two constitutive *Ras* constructs used in this paper (*Ras^V12^(2)*) with or without *RalA* knockdown within stem/progenitor cells (ISC/EB>; green). (**D**) Quantification of average pERK staining intensity as seen in (**C**) within the ISC/EB compartment (GFP positive). Two-way ANOVA followed by Sidak's multiple comparisons test; n = number of z-stack confocal images quantified, each from an independent posterior midgut. (**E**) Representative confocal images of pH3 staining (red) within the ISC/EB compartment (green) of control animals or animals overexpressing two types of constitutively active *Egfr* constructs (*EGFR^λtop^* or *EGFR^A887T^*) with or without *RalA* knockdown within stem/progenitor cells (ISC/EB>; green). Scale bar = 50 µm. (**F**) Quantification of pH3-positive nuclei in posterior midguts as in (**E**). Two-way ANOVA followed by Sidak's multiple comparisons test. Error bars represent SEM. n = number of midguts. (**G**) Representative confocal images of pERK staining (red/grey) within the ISC/EB compartment (green) of control animals or animals overexpressing two types of constitutively active *Egfr* constructs (*EGFR^λtop^* or *EGFR^A887T^*) with or without *RalA* knockdown within stem/progenitor cells (ISC/EB>; green). (**H**) Quantification of average pERK staining intensity as in (**G**) within the ISC/EB compartment (GFP positive). Two-way ANOVA followed by Sidak's multiple comparisons test; n = number of z-stack confocal images quantified, each from an independent posterior midgut. Where indicated: *p<0.05, **p<0.01, ***p<0.001, ****p<0.0001, ns: not significant. All error bars represent SD. Scale bars = 20 µm, unless otherwise stated.

The online version of this article includes the following source data and figure supplement(s) for figure 3:

**Source data 1.** Ral GTPases are required for EGFR/MAPK signalling upstream of Ras.
**Figure supplement 1.** Ral GTPases are required for EGFR/MAPK signalling upstream of Ras.
**Figure supplement 1—source data 1.** Ral GTPases are required for EGFR/MAPK signalling upstream of Ras.

Consequently, oncogenic Ras, whose activation is independent of EGFR signalling, is refractory to RalA function in the intestine (*Figure 3A–D* and *Figure 3—figure supplement 1C, D*).

Next, we used a surface biotinylation-based biochemical assay to directly quantify the rate of EGFR internalisation in H1299, a human non-small cell lung cancer (NSCLC) cell line with intact EGFR signalling (*Amann et al., 2005*). To obtain a measure of endocytosis that was not influenced by the rate at which the receptor returns, or 'recycles', to the cell surface from endosomes, we performed the surface biotinylation-based assay in the presence of the receptor recycling inhibitor, primaquine. This clearly indicated that EGF-driven (but not EGF-independent) endocytosis of EGFR was significantly reduced by combined knockdown of *Rala* and *Ralb* (*Figure 4G* and *Figure 4—figure supplement 3A, B*). By contrast, integrin α5β1, transferrin (hTfnR), or ligand-induced c-Met receptor internalisation were not affected by *Rala/b* knockdown (*Figure 4—figure supplement 3C–F*). These

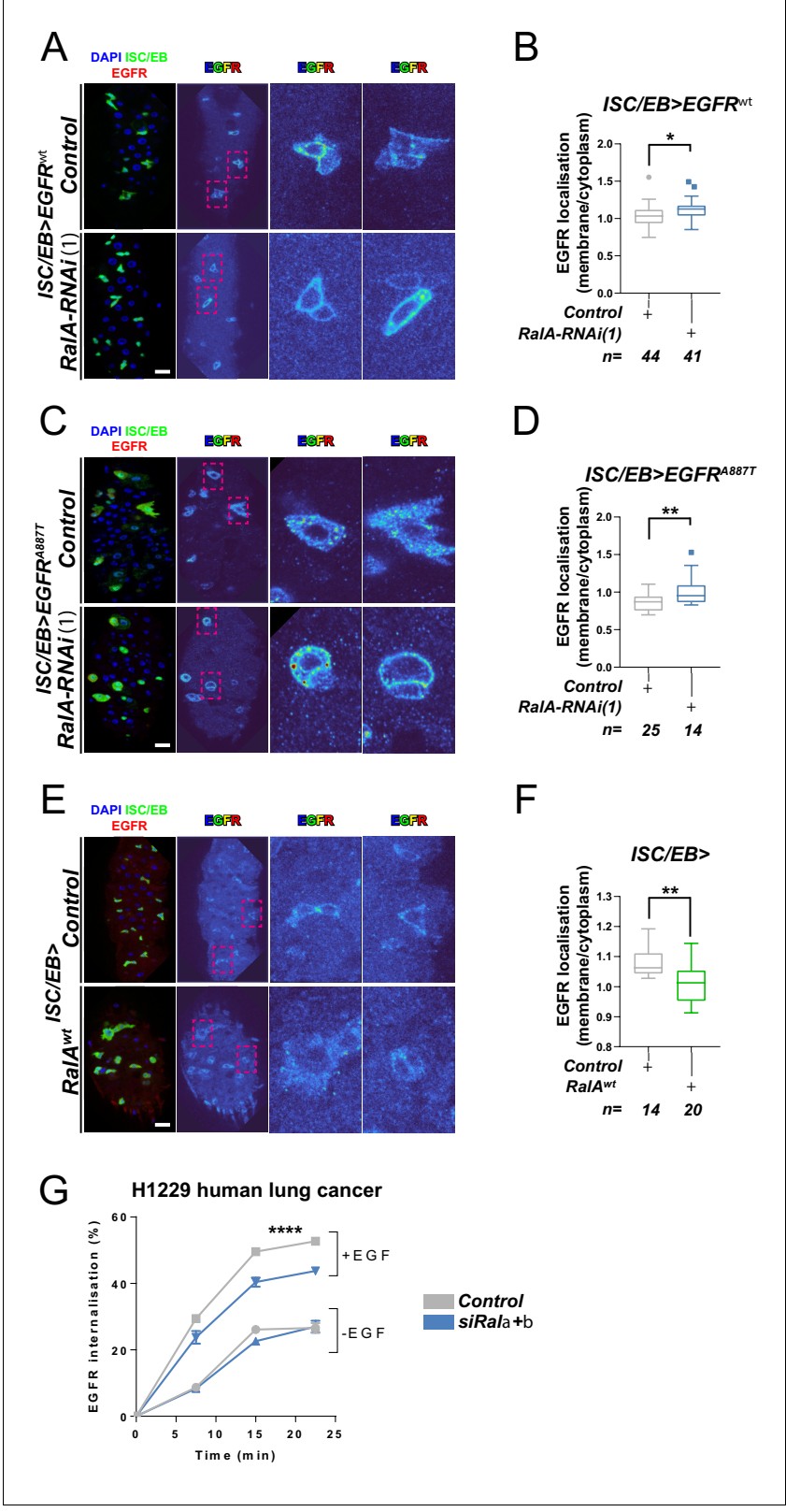

**Figure 4.** Ral GTPases are required for EGFR internalisation. (**A**) Representative images of wild-type EGFR staining (red/turbo colour map) in adult *Drosophila* midgut stem/progenitor cells (intestinal stem cell/enteroblast [ISC/EB>]; green) without (Control) or with *RalA* knockdown (*RalA-RNAi*). (**B**) Quantification of EGFR plasma membrane staining localisation in midguts as in (**A**) relative to the cytoplasm. Data is presented as Tukey's box and whiskers

*Figure 4 continued*

plot. Data were analysed by Student's t-test. n = number of z-stack confocal images quantified, each from an independent posterior midgut. (C) Representative images of EGFR$^{A887T}$ staining (red/turbo colour map) in adult *Drosophila* midgut stem/progenitor cells (ISC/EB>; green) without (Control) or with *RalA* knockdown (*RalA-RNAi*). (D) Quantification of EGFR$^{A887T}$plasma membrane staining localisation as in (C) relative to the cytoplasm presented as Tukey's box and whiskers plot. Student's t-test. n = number of z-stack confocal images quantified, each from an independent posterior midgut. (E) Representative images of EGFR staining in (red/turbo colour map) in adult *Drosophila* midgut stem/progenitor cells (ISC/EB>; green) without (Control) or with wild-type *RalA* overexpression (*RalA$^{wt}$*). (F) Quantification of EGFR plasma membrane staining localisation in midguts as in (E) relative to the cytoplasm. Data is presented as Tukey's box and whiskers plot. Student's t-test. n = number of z-stack confocal images quantified, each from an independent posterior midgut. (G) Internalisation of EGFR over time determined by a surface biotinylation ELISA-based assay in H1299 human non-small cell lung cancer cells transfected with a non-targeting (*Control*) or combined *Rala* and *Ralb* knockdown constructs (si*Rala +b*) and incubated in the presence or absence of EGF ligand. Data from one experiment with three technical replicates and representative of three independently performed experiments is presented. Two-way ANOVA followed by Bonferroni's multiple comparisons test. Error bars represent SEM. Where indicated: *p<0.05, **p<0.01, ***p<0.001, ****p<0.0001. All error bars represent SD. Scale bars = 20 μm.

The online version of this article includes the following source data and figure supplement(s) for figure 4:

**Source data 1.** Ral GTPases are required for EGFR internalisation.
**Figure supplement 1.** Demonstration of method used to quantify EGFR cellular localisation.
**Figure supplement 2.** *Ras* is required for EGFR internalisation.
**Figure supplement 2—source data 1.** *Ras*is required for EGFR internalisation.
**Figure supplement 3.** RAL GTPases are required for EGFR internalisation.
**Figure supplement 3—source data 1.** RAL GTPases are required for EGFR internalisation.

results suggest that the effect of RAL GTPases on EGFR cellular localisation is conserved between *Drosophila* and mammals, and that RAL proteins function in a context-dependent manner, as opposed to being generally required for transmembrane or tyrosine kinase receptor trafficking dynamics.

## RAL proteins are necessary for EGFR-dependent tumorigenesis

Given that intestinal hyperplasia caused by hyperactivation of β-catenin or oncogenic RAS is independent of RAL proteins (*Johansson et al., 2019*; *Figure 3A, B* and *Figure 3—figure supplement 1C, D*), the importance of RAL GTPases in intestinal malignancy remains unaddressed. The effect of *RalA* knockdown on intestinal hyperproliferation caused by overexpression of wild-type or constitutively active mutants of EGFR in the intestine (*Figure 3*) suggests that other pathological settings driven by exacerbated EGFR activity might also be sensitive to RAL function.

c-Src is a conserved non-receptor tyrosine kinase whose expression is necessary and sufficient to drive regeneration and tumourigenesis of both the *Drosophila* and mouse intestine through EGFR/ MAPK activation (*Cordero et al., 2014*; *Kohlmaier et al., 2015*; *Figure 5A, B*). Consistently, *Src* overexpression in ISCs/EBs (*esg$^{ts}$>Src64*wt)-induced expression of the MAPK pathway transcriptional target Sox21a (*Figure 5C, D*) and pERK levels (*Figure 5E, F*; *Cordero et al., 2014*; *Kohlmaier et al., 2015*). Importantly, knocking down *RalA* (*ISC/EB>Src64$^{wt}$; RalA-RNAi*) suppressed Src-driven ISC hyperproliferation and MAPK signalling activation in the *Drosophila* midgut (*Figure 5A-F*), which correlated with an increase in membrane versus intracellular levels of EGFR in *ISC/EB>Src64$^{wt}$; RalA-RNAi* midguts when compared to *ISC/wt>Src64*wt counterparts (*Figure 5G, H*).

As a proof of principle in an orthogonal mammalian system dependent on EGFR for morphogenesis, we employed the human breast tumour cell line HMT3522 T4-2 (henceforth referred to as 'T4-2') as a paradigm to test the role of mammalian RAL GTPases in malignant growth. T4-2 is a subline obtained after spontaneous malignant transformation of the benign breast tumour cell line HMT3522 S1 (henceforth 'S1'). Compared to the S1 predecessor, T4-2 cells grow as disorganised aggregates of cells when cultured in 3D extracellular matrix gels such as Matrigel. This growth and morphogenesis in 3D of T4-2 cells is EGFR-dependent: T4-2 show robustly upregulated EGFR levels and activation, their growth is independent of exogenous EGF, and they are acutely sensitive to

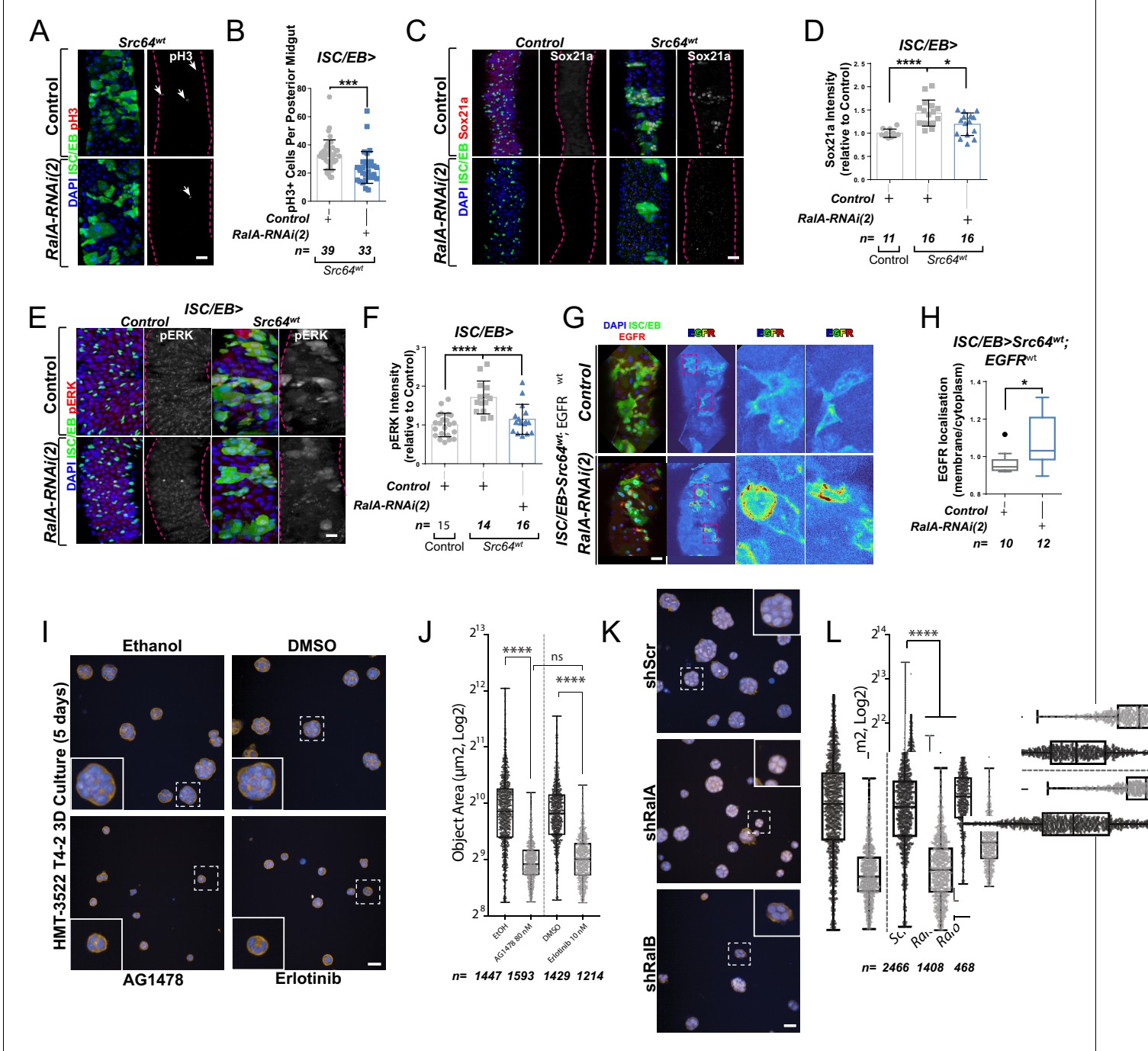

**Figure 5.** Ral GTPases mediate malignant transformation of the intestinal and mammary epithelium. (**A**) Representative confocal images of pH3 staining (red/grey) in midguts overexpressing *Src*-kinase (*Src64*[wt]) with or without *Rala* knockdown (*RalA-RNAi(2)*) in stem/progenitor cells (intestinal stem cell/ enteroblast [ISC/EB>]; green). White arrows indicate pH3-positive nuclei. (**B**) Quantification of pH3-positive nuclei in posterior midguts as in (**A**). Data were analysed by Student's t-test. n = number of midguts. (**C**) Representative confocal images of Sox21a staining (red/grey) in midguts overexpressing *Src*-kinase (*Src64*[wt]) with or without *Rala* knockdown (*RalA-RNAi(2)*) in stem/progenitor cells (ISC/EB>; green). Scale bar = 50 µm. (**D**) Quantification of average Sox21a staining intensity within the nuclear compartment (DAPI positive) as in (**C**). Two-way ANOVA followed by Sidak's multiple comparisons test; n = number of z-stack confocal images quantified, each from an independent posterior midgut. (**E**) Representative confocal images of pERK staining (red/grey) in midguts overexpressing *Src*-kinase (*Src64*[wt]) with or without *Rala* knockdown (*RalA-RNAi(2)*) in stem/progenitor cells (ISC/EB>; green). (**F**) Quantification of average pERK staining intensity within the ISC/EB compartment (GFP positive) as in (**E**). Two-way ANOVA followed by Sidak's multiple comparisons test; n = number of z-stack confocal images quantified, each from an independent posterior midgut. Error bars represent SD. (**G**) Representative images of EGFR staining (red/grey) in midguts overexpressing *Src*-kinase (*Src64*[wt]) and *EGFR*[wt] with or without *Rala* knockdown

*Figure 5 continued on next page*

*Figure 5 continued*

(*RalA-RNAi(2)*)) in stem/progenitor cells (ISC/EB>; green). (H) Quantification of EGFR plasma membrane staining localisation relative to the cytoplasm as in (G) presented as Tukey's box and whiskers plot. Data were analysed by Student's t-test. n = number of z-stack confocal images quantified, each from an independent posterior midgut. (I) Confocal fluorescence microscopy images of HMT3522 T4-2 3D cultures, treated with EGFR inhibitors (tyrphostin AG1478 and erlotinib) or corresponding vehicle controls (ethanol and DMSO, respectively) followed by fixation after 5 days and staining for F-actin (yellow) and nuclei (blue, Hoechst). Scale bar = 40 μm. (J) Quantification of area of 5 days T4-2 cysts treated as in (I). n ≥ 1214 cysts assessed from four wells/condition/experiment, two independent experiments. One-way ANOVA, Tukey's multiple comparisons test. (K) Confocal fluorescence microscopy images of HMT3522 T4-2 cysts of 5 days expressing either scramble, RalA or RalB shRNA. Cysts were fixed and stained for F-actin (yellow) and nuclei (blue, Hoechst). Scale bar = 40 μm. (L) Quantification of 5 days T4-2 cysts as in (K). n ≥ 468 cysts assessed from four wells/condition/experiment, three independent experiments. One-way ANOVA, Tukey's multiple comparisons test. Where indicated: *p<0.05, **p<0.01, ***p<0.001, ****p<0.0001, ns: not significant. All error bars represent SD. Scale bars = 20 μm, unless otherwise stated.

The online version of this article includes the following source data and figure supplement(s) for figure 5:

**Source data 1.** Ral GTPases mediate malignant transformation of the intestinal and mammary epithelium.

**Figure supplement 1.** *Ral* knockdown in human mammary cell lines.

**Figure supplement 1—source data 1.** *Ral*knockdown in human mammary cell lines.

EGFR inhibitors (*Madsen et al., 1992*; *Wang et al., 1998*). Thus, we hypothesised that T4-2 growth would be dependent on RAL function.

Consistent with previous reports (*Madsen et al., 1992*; *Wang et al., 1998*), treating T4-2 cells with two structurally independent EGFR inhibitors, tyrphostin (AG1478) and erlotinib, resulted in defective growth as determined by a reduction in 3D acinus size (*Figure 5I, J*). Importantly, stable depletion of *Rala* or *Ralb* in T4-2 by shRNA (*Figure 5K, L* and *Figure 5—figure supplement 1A, B*) phenocopied EGFR inhibition, as determined by a significant reduction in 3D acinus size (*Figure 5K, L*). Therefore, RALA/B function is similarly required for a mammalian morphogenetic function that is dependent on EGFR. Altogether, our results uncover a conserved role of RAL GTPases mediating EGFR/MAPK-dependent tissue homeostasis and transformation.

## Discussion

Spatial and temporal regulation of signal transduction by the endocytic pathway plays a key role in health and pathophysiology (*Casaletto and McClatchey, 2012*; *von Zastrow and Sorkin, 2007*). The impact of this process in adult stem cells and tissue homeostasis is only recently becoming evident from reports on the effect of endocytosis and autophagy on ISC proliferation through modulation of Wnt/β-catenin and EGFR/MAPK activity, respectively (*Johansson et al., 2019*; *Zhang et al., 2019*).

In this study, we identify a role for the Ras-related protein RAL in the activation of EGFR/MAPK signalling activity through regulation of EGFR internalisation (*Figure 6*). Preventing RAL function in *Drosophila* intestinal stem/progenitor cells reduces the intracellular pool of EGFR, leading to decreased MAPK activation and downstream signalling. This role of RAL proteins impacts stem cell proliferation and regeneration of the intestinal epithelium and has implications in pathological settings that depend on active EGFR signalling, including intestinal hyperplasia and breast cancer cell growth. However, oncogenic Ras expression in the intestine escapes the antiproliferative effect of *Ral* knockdown.

### RAL GTPases as regulators of signal transduction

While internalisation is recognised as the initial means to attenuate signal transduction through reduction of PM receptors available for activation by extracellular ligands (*Goh et al., 2010*; *Sousa et al., 2012*; *Vieira et al., 1996*; *von Zastrow, 2003*), the subsequent outcome of endocytosis on signalling is dependent on the trafficking pathway followed by internalised receptors. Internalisation of membrane EGFR through Clathrin-mediated endocytosis results in prolonged EGFR signalling by favouring receptor recycling back to the PM, while Clathrin-independent endocytosis leads to EGFR degradation and signalling attenuation (*Sigismund et al., 2008*). The differential effect of endocytic trafficking on EGFR has therapeutic implications as Clathrin inhibition can divert a tyrosine kinase inhibitor (TKI)-resistant form of EGFR from Clathrin-mediated endocytosis and recycling to pinocytosis and degradation in non-small cell lung carcinoma (*Ménard et al., 2018*).

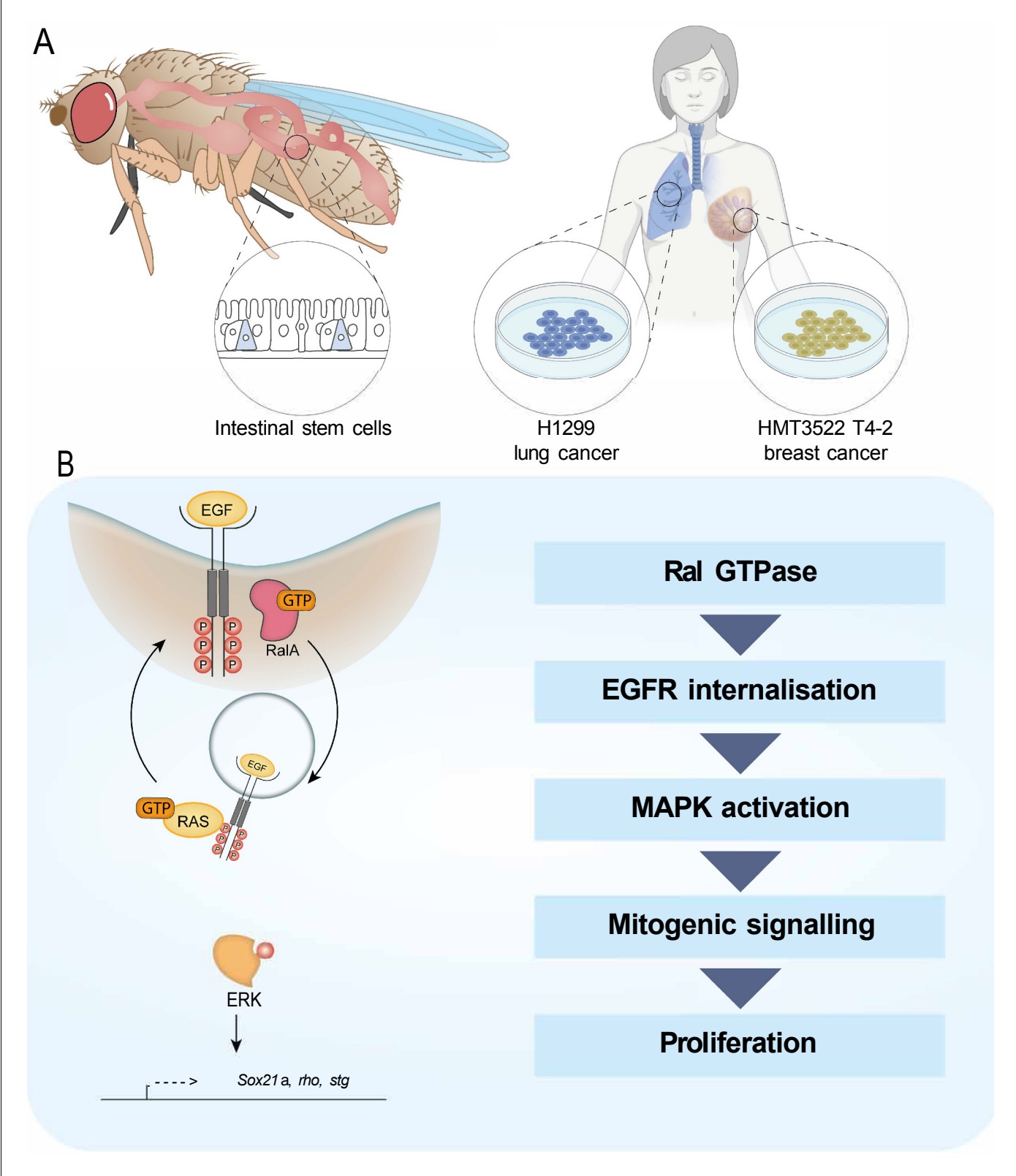

**Figure 6.** Working model depicting the role of RAL GTPases in EGFR/MAPK signalling. (**A**) Experimental contexts used. Most results were acquired from *Drosophila* intestinal epithelial stem progenitor cells. Key findings were confirmed using mammalian intestine and human lung and breast cancer cell lines. (**B**) RalA is necessary for EGFR internalisation and MAPK activation, leading to mitogenic signalling.

Here, we provide robust evidence of physiological and pathological contexts in the intestine where the internalisation of EGFR mediated by RAL GTPases directly correlates with potentiation of downstream MAPK signalling (*Figure 6*). We recently reported a similar effect of RAL proteins on the seven transmembrane class receptor, Frizzled, leading to high threshold of Wnt signalling activity (*Johansson et al., 2019*). In both cases, the ultimate outcome of RAL action is an efficient acute proliferative response of intestinal stem cells during tissue regeneration following damage. Therefore, RAL GTPases are effectors of two pivotal signal transduction pathways within the intestinal epithelium (*Biteau and Jasper, 2011*; *Buchon et al., 2010*; *Jardé et al., 2020*; *Jiang et al., 2011*; *Perochon et al., 2018*; *Sato et al., 2009*; *Xu et al., 2011*). The effect of knocking down *RalA* in the *Drosophila* midgut is, however, milder than that observed upon individual or combined impairment of Wnt/β-catenin and EGFR/MAPK signalling reception in ISCs (*Xu et al., 2011*). This suggests that RalA is only partly responsible for the activation of these signalling pathways and its effect is only evident in the regenerative response to damage, which requires high thresholds of signalling activity to allow acute stem cell proliferation for tissue regeneration. The scenario is different in the mammalian intestine, where combined knockout of *Rala* and *Ralb* leads to complete disruption of intestinal epithelial homeostasis (*Johansson et al., 2019*). This may relate to inherent differences in the signalling activity levels needed to maintain homeostatic ISC proliferation in the fly midgut versus the mouse intestine. Compared to its murine counterpart, basal proliferation in the adult fly midgut is relatively low and there is no transit-amplifying proliferative zone (*Micchelli and Perrimon, 2006*; *Ohlstein and Spradling, 2006*). Alternatively, the difference could lie in the different experimental approaches taken, namely the use of gene knockout in the mouse versus partial knockdown in the fly. Our efforts to generate FRT-mediated *Rala* knockout clones in the adult *Drosophila* midgut were unsuccessful (data not shown), and full mutant animals are not viable. Therefore, any potential residual activity due to incomplete knockdown could lead to milder *Drosophila* phenotypes.

RAL GTPases have been linked to Clathrin-mediated endocytosis via interaction of their effector protein, RAL binding protein (RALBP1), with the Clathrin adaptor AP2 (*Jullien-Flores et al., 2000*). More recently, RAL proteins have also been shown to engage in Caveolin-mediated endocytosis (*Jiang et al., 2016*). While the potentiating effect of RALs on EGFR signalling activity would favour a role of the small GTPases in Clathrin-mediated endocytosis in the system, this needs to be directly assessed. Experiments to functionally connect RalA with specific endocytic trafficking pathways using *Drosophila* genetics have been unsuccessful as, consistent with recently published work (*Zhang et al., 2019*), global perturbation of the trafficking machinery within ISCs leads to very severe disruption of intestinal homeostasis (data not shown), precluding the establishment of meaningful genetic interactions.

Future research will need to be done to better elucidate the place of action of RAL GTPases within the endocytic trafficking pathway and its connection with EGF and Wnt receptors in the intestine. The use of fluorescently tagged endocytic proteins (*Dunst et al., 2015*) combined with recently developed live imaging approaches in the adult *Drosophila* intestine (*Koyama et al., 2020*; *Martin et al., 2018*) offers a clear opportunity to visualise spatial and temporal receptor/endosome interactions in vivo.

## RAL GTPases as potential therapeutic targets in cancer

EGFR function is frequently altered in cancer (*Santarius et al., 2010*; *Yarden and Pines, 2012*). Excessive protein levels due to gene amplification or increased-transcription are the most common EGFR perturbations found in gastrointestinal and lung adenocarcinoma as well as in cholangiocarcinoma (*Birkman et al., 2016*; *Jung et al., 2017*; *Li et al., 2008*). On the other hand, EGFR kinase domain activating point mutations are associated with non-small cell lung carcinoma and glioblastoma, but are rarely seen in other types of cancer (*Li et al., 2008*; *Siegelin and Borczuk, 2014*; *Zhang et al., 2016*). Extracellular domain truncating mutations yielding to constitutively active receptor through ligand-independent dimerisation have also been observed in glioblastomas (*Furnari et al., 2015*; *Guo et al., 2015*; *Huang et al., 1997*). We have utilised *Drosophila* genetic constructs that mimic all three main classes of EGFR common to human cancers and which lead to intestinal hyperplasia when overexpressed in intestinal stem/progenitor cells (*Figure 3*). Genetic inhibition of Ral GTPase activity consistently prevented hyperproliferation in these models, suggesting that targeting RAL function could be a potentially effective therapeutic approach in the treatment of multiple highly aggressive cancer types.

Current EGFR-targeted therapies include small-molecule TKIs and monoclonal antibodies (mAbs) against the extracellular domain of the receptor (*Xu et al., 2017*). A number of resistance mechanisms arise secondary to treatment. Specific kinase domain mutations desensitise cells against TKI (*Sequist et al., 2011*; *Yu et al., 2015*), while alterations of the antibody binding site are observed in CRC (*Arena et al., 2015*). There is also a tendency for downstream mutations (Raf, Ras, MAPK, MET) to uncouple pathway activity from the receptor (*Camidge et al., 2014*; *Mancini and Yarden, 2016*). The most common form of resistance to EGFR-targeted therapies is believed to be innate rather than adaptive (*Parseghian et al., 2019*). Indeed, about 80% of CRCs are refractory to EGFR therapy (*Bardelli and Siena, 2010*). Several reports highlight how cancer cells co-opt the endocytic pathway for growth and survival benefits (*Mosesson et al., 2008*). In fact, these have been proposed as a potential venue for drug development (*Mellman and Yarden, 2013*). However, based on the current evidence, we propose that targeting RAL function versus a broader component of the endocytic machinery may prove a more refined approach, leading to lower toxic effects (*Zhang et al., 2019*).

RAL effector proteins, including RALGEFs and RALBP1, have emerged as important mediators of malignant growth in pancreatic, colorectal, prostate, bladder, and other tumour cell lines characterised by the presence of oncogenic *RAS* mutations (*Neel et al., 2011*). Furthermore, genetic knockout of the RALGEF, RALGDS, ameliorates tumour growth in a mouse model of Ras-driven skin tumourigenesis (*González-García et al., 2005*). Unexpectedly, our results show that, at least in the intestine, oncogenic mutations in *Ras* are refractory to Ral GTPase inhibition. These apparently discrepant results could be due to context-dependent requirements for RAL function in malignancy, differences between in vitro and in vivo experimental settings, and/or a potential promiscuous role of RAL effectors on small GTPase signalling.

Previously, we have shown that fly and murine intestines bearing loss of *Apc*, a key initiating event in up to 80% of human CRC, also overcome the need for RAL GTPases to proliferate (*Johansson et al., 2019*). Taken together, our results argue against an effective role of anti-RAL therapies to treat CRCs carrying *Apc* loss-of-function and/or hyperactivating *Ras* mutations. On the other hand, tumours with overexpression or activating mutations in EGFR, such as carcinomas of the upper gastrointestinal tract, lung and mammary tissue, or glioblastomas (*Birkman et al., 2016*; *Furnari et al., 2015*; *Guo et al., 2015*; *Huang et al., 1997*; *Li et al., 2008*; *Siegelin and Borczuk, 2014*; *Zhang et al., 2016*), might be responsive to impairment of RAL function. Ultimately, taking into consideration the genetic composition of the tumour is of outmost importance when considering the use of RAL inhibition as a therapeutic approach.

## Materials and methods

Key resources table is included as Appendix 1.

### Experimental models and organisms

#### Species used

*Drosophila melanogaster* and *Mus musculus*.

Only mated females were used for *Drosophila* experiments.

#### Cell lines

HMT3522 T4-2 (human breast cancer derived; from Valerie Weaver), NCI-H1299 (human lung cancer derived; from ATCC), HEK293-FT (human kidney derived; from Thermo Fisher Scientific). All cell lines used in this study were authenticated through STR profiling using Promega Geneprint 10 Kit. Gene fragment analysis was performed on a 3130xl Genetic Analyser, and Genemapper v5 was used for analysis. Cell lines were confirmed negative for mycoplasma.

#### *Drosophila* breeding and maintenance

Flies were maintained in humidity and temperature-controlled incubators with a 12–12 hr light-dark cycle. Crosses were kept at 18°C. F1s of the desired genotype were collected 2–3 days after adult eclosion and aged at 29°C for the time needed to allow for transgene activation. Only female

midguts were used. Standard rearing medium used 10 g agar, 15 g sucrose, 30 g glucose, 15 g maize meal, 10 g wheat germ, 30 g treacle, and 10 g soya flour per litre of distilled water.

Exact genotypes for all figure panels are listed in *Supplementary file 1*.

## Mouse work

Mouse experiments were performed as described in *Johansson et al., 2019* according to the UK Home Office regulations and designed in accordance with the ARRIVE guidelines. Animals were fed on standard diet and water ad libitum, and under non-barrier conditions. Genotypes used are indicated in the Key resources table. *Vil1CreER* recombinase was induced using 80 mg/kg Tamoxifen (Sigma) IP. Regeneration was induced using caesium-137 γ-radiation sources delivering 0.423 Gy min$^{-1}$ to a total of 10 Gy. Mice were sampled 3 days following irradiation damage. No distinction was made between males and females in the mouse experiments. All animals used in experiments were above 20 g of weight. Experiments were performed on a C57BL/6 background and using a minimum of three mice per condition/genotype.

## IHC of mouse tissue

Formalin-fixed paraffin-embedded (FFPE) tissues were cut into 4 µm sections and mounted onto adhesive slides, followed by a 2-hr-long oven-incubation step at 60°C. Samples were dewaxed in xylene for 5 min before rehydration through serial washes in decreasing concentrations of alcohol followed by washing with H$_2$O for 5 min. For heat-induced epitope, retrieval sections were heated for 20 min at 97°C in sodium citrate pH6 retrieval buffer (Thermo, TA-250-PM1X) before cooling to 65°C. This was followed by washing in Tris Buffered Saline with Tween (TBT) (Thermo, TA-999-TT). Sections were loaded onto the Dako autostainer link48 platform, washed with TBT, then peroxidase blocking solution (Agilent, S2023) for 5 min. Sections were washed with TBT, then appropriate antibody was applied to specific slides. Phospho-p44/42 MAPK (Erk1/2) (Cell Signalling, 9101) was applied at 1/400 dilution, and p44/42 MAPK (Erk1/2) (Cell Signalling 9102) was applied at 1/40 dilution for 30 min. After another TBT wash, secondary antibody (Rabbit Envision, Agilent, K4003) was applied for 30 min before washing with TBT again. 3,3′ diaminobenzidine (Agilent, K3468) was then applied for 10 min before washing in H$_2$O to terminate the reaction. Finally, slides were counterstained with haematoxylin and dehydrated in increasing concentrations of alcohol, then taken through three changes of xylene prior to sealing with glass coverslips using DPX mounting media for microscopy.

## Quantification of pERK and total ERK staining in mouse tissues

A minimum of 12 and up to 30 randomly selected crypts per animal from at least three mice per genotype, per condition were quantified. Data are expressed as the percentage of crypt cells positively stained for a marker of interest per crypt. Finally, the percentage of positively stained cells was averaged for each animal.

## Brightfield microscopy and scoring of adult wing patterning

*Drosophila* wings were mounted onto glass slides (VWR) with 13 mm × 0.12 mm spacers (Electron Microscopy Science). Images were obtained on the ZEISS Axio Observer system. Images were focus stacked using the ZEN 2 software (ZEISS). Wing dysmorphia was blindly scored on a scale from 1 to 5 using a previously developed macro https://github.com/emltwc/TracheaProject/blob/master/Blind_scoring.ijm (copy archived at swh:1:rev:2ef7574e3c9bbb7ef852655511a86ef7531d35bb); *Naszai, 2021a*, where 1 is a normal, wild-type wing and 5 refers to the most severely disrupted adult wings.

## Immunofluorescence of *Drosophila* tissues

Immunofluorescent staining was performed as described in *Johansson et al., 2019*. Briefly, tissues were dissected in PBS and immediately fixed in 4% paraformaldehyde (PFA; Polysciences Inc) at room temperature for a minimum of 30 min. Once fixed, 20-min-long washes in PBS + 0.2% Triton X-100 (PBST) were repeated three times, followed by overnight incubation at 4°C with primary antibodies in PBST + 0.5% bovine serum albumin (BSA) (PBT). Prior to applying the secondary

antibodies, tissues were washed in PBST three times 20 min and then incubated with the appropriate antibodies in PBT for 3 hr at room temperature, followed by washing and mounting.

Midguts stained for pERK and tERK included a methanol fixation step between PFA fixation and PBST washing steps of the standard protocol. Following PFA, fixation methanol was added dropwise to the solution, with the tissues in it until the volume of the liquid is at least double. Tissues were transferred into 100% methanol for minimum 1 min. PBS was added to the methanol dropwise to rehydrate the tissues after which the samples were subjected to the standard staining protocol.

All samples were mounted onto glass slides (VWR) with 13 mm $\times$ 0.12 mm spacers (Electron Microscopy Science) and VECTASHIELD antifade mounting medium containing DAPI (Vector Laboratories, Inc). Confocal images were obtained on a ZEISS LSM 780 and processed in the ZEISS ZEN software.

Antibody concentrations used were as follows: anti-GFP (1:2000), anti-pERK (1:100), anti-tERK (1:100), anti-EGFR (1:50), anti-Sox21a (1:2000), and anti-pH3S10 (1:100). Secondary antibodies were used as follows: anti-chicken-IgY-488 (1:200), anti-rabbit-IgG-594 (1:100), and anti-mouse-IgG-594 (1:100).

### *Drosophila* midgut regeneration assay

*Drosophila* intestinal regeneration was induced through oral infection using *Erwinia carotovora* subsp. *carotovora 15* (*Ecc15*) (*Basset et al., 2000*), as described in *Neyen et al., 2014*. Briefly, bacteria were cultured overnight in LB medium in an orbital shaker incubator at 29°C, 200 rpm. Samples were pelleted (Beckman Coulter JS-4.2 rotor, 10 min @3000 rpm) and adjusted to $OD_{600}$ = 200. Flies used for regeneration experiments were starved in empty vials for 2 hr prior to infection to synchronise feeding. Animals were moved into vials containing filter paper (Whatman) soaked into vehicle control, 5% sucrose solution (Mock), or the prepared bacterial solution mixed with 5% sucrose 1:1. Flies were dissected 12–16 hr after infection.

### Staining quantification

pERK and tERK intensity were quantified in 16-bit z-stack confocal images as the average staining intensity within the GFP-positive compartment. Sox21a staining was quantified in 16-bit z-stack confocal images as the average staining intensity within the entire DAPI-positive compartment. pERK, tERK, and Sox21a were quantified using the custom ImageJ macro: BatchQuantify (https://github.com/emltwc/2018-Cell-Stem-Cell, copy archived at swh:1:rev:e45f961ed6217ecc0bece566a76a633fd2b47ec0), *Naszai, 2021b*. EGFR membrane/cytoplasmic staining was quantified in 16-bit z-stack confocal images using the custom ImageJ macro: EGFR_quant (https://github.com/emltwc/EGFRProject, (copy archived at swh:1:rev:4888f27a6766694b33a8b25bcb42a078fa786f8d)).

### Survival quantification

Relative survival was calculated by counting the proportion of adult flies emerging from crosses, which carried the desired experimental genotypes, as per the expected Mendelian ratio. When the proportion of animals of a given genotype emerged at the expected Mendelian ratio, this genotype was deemed to be 100% viable.

### *Drosophila* RNA extraction, RNA-sequencing, and RT-qPCR

Total RNA from a minimum of 25 midguts was extracted using QIAGEN RNAeasy kit, following the manufacturer's instructions, including the on-column DNase digestion step. For RNA-seq, an RNA integrity score was determined (average = 9.4, SD = 0.6, lowest score used = 8.2; Agilent Technologies 2200 Tapestation, RNA Screen Tape). Libraries for cluster generation and DNA sequencing were prepared following *Fisher et al., 2011* using Illumina TruSeq RNA library Preparation Kit v2. Libraries were run on the Next Seq 500 platform (Illumina) using the High Output 75 cycles kit (2 $\times$ 36 cycles, paired end reads, single index).

For RT-qPCR, RNA was quantified using a NanoDrop 2000c Spectrophotometer. cDNA was synthesised using the High-Capacity cDNA reverse transcription kit (Applied Biosystems) according to the manufacturer's recommendations using a maximum of 2 μg RNA per 20 μL final volume. Quanta SYBR Green Master Mix (Low ROX, Fermentas) was used following the manufacturer's instructions.

Data were obtained and analysed using the Applied Biosystems 7500 software. Results represent four independent replicates ± SEM. Expression of target genes was measured and normalised to *rpl32* using standard curves.

## Western blot

Protein was extracted from 12 adult female *Drosophila* midguts dissected in ice-cold PBS. The tissues were lysed in 20 µL RIPA buffer (Sigma) using a microcentrifuge pestle. Samples were spun down at 13,000 rpm for 10 min at 4°C and the supernatant was collected. Protein concentration was determined using Bradford reaction (Abcam) following the manufacturer's recommendations. 40 µg of total protein was loaded onto NuPAGE 10% Bis-Tris gel (Thermo Fisher Scientific) and run using NuPAGE MOPS buffer (Thermo Fisher Scientific). Protein was transferred to a membrane (Bio-Rad) using the Trans-Blot Turbo system (Bio-Rad) following the manufacturer's instructions. Membranes were blocked overnight at 4°C in 5% BSA (Sigma), then probed using pERK and tERK antibodies (Cell Signalling) at 1:1000 concentration. Antibody signal was detected using the SuperSignal West Pico Chemiluminescent Substrate (Thermo Fisher Scientific) system.

## Cell culture

HMT3522 T4-2 (V. Weaver, UCSF) cells were cultured in precoated collagen plates using DMEM/Ham's F12 (1:1) medium supplemented with 2 mM glutamine (Life Technologies), 250 ng/mL insulin solution from bovine pancreas (Sigma-Aldrich), 10 µg/mL transferrin (Sigma-Aldrich), 2.6 ng/mL sodium selenite (Sigma-Aldrich), $10^{-10}$ M 17 β-estradiol (Sigma-Aldrich), $1.4 \times 10^{-6}$ M hydrocortisone (Sigma-Aldrich), and 10 ng/mL human prolactin (Miltenyi Biotec).

3D acini were grown as follows: single-cell suspensions ($1.5 \times 10^4$ cells per mL) were plated in the appropriate medium supplemented with 2% Growth Factor Reduced Matrigel (GFRM; BD Biosciences). 100 µL of this mix were added per well in a 96-well ImageLock plate (Essen Biosciences) precoated with 10 µL of pure GFRM for 15 min at 37°C. Cells were incubated at 37°C for 5 days, changing the media every two days, before IF.

For inhibitor studies, cells were treated from the time of plating with Tyrphostin-AG1478 (80 nM in ethanol, Sigma-Aldrich), erlotinib (100 nM in DMSO), and ethanol or DMSO as appropriate controls, respectively.

HEK293-FT (Thermo Fisher Scientific) were cultured in DMEM supplemented with 10% FBS, 6 mM L-glutamine, and 0.1 mM non-essential amino acids (NEAA) (all reagents from Life Technologies/Thermo Fisher).

## Generation of stable cell lines

Stable cell lines were performed by co-transfecting HEK293-FT packaging cells with a pLKO.1-puromycin shRNA plasmid with VSVG and SPAX2 lentiviral packaging vectors using Lipofectamine 2000 according to the manufacturer's instructions (Invitrogen). Viral supernatants were collected, filtered using PES 0.45 µm syringe filters (Starlab), and concentrated using Lenti-X Concentrator (Clontech) as per the manufacturer's instructions. Cells were then transduced with the lentivirus for 3 days before selection with 1 µg/mL puromycin (Thermo Fisher Scientific). shRNA target sequences: non-targeting control shScr (5'CCGCAGGTATGCACGCGT3'), shRalA (5'GGAGGAAGTCCAGATCGATAT3'), and shRalB (5'CAAGGTGTTCTTTGACCTAAT3'). To knockdown RAL protein expression in H1229 cells, cells were transfected with Dharmacon ON-TARGETplus siRNAs using the Amaxa Nucleofector system (Lonza).

## RNA extraction and RT-qPCR in cell culture samples

RT-qPCR on human samples was performed following the same protocol used for *Drosophila* samples, except using human β-actin or GAPDH to normalise transcript levels using the delta-delta-$C_T$ method.

## Cyst growth assay

Acini labelling was adapted from previously described protocols. Briefly, cultures were fixed in 4% PFA (Affymetrix) for 10 min at room temperature (RT), washed twice in PBS, blocked for 1 hr in PFS buffer (PBS, 0.7% w/v fish skin gelatin; Sigma-Aldrich), 0.5% saponin (Sigma-Aldrich), and incubated

with primary antibodies diluted in PFS at 4℃ overnight with gentle rocking. Then, cyst cultures were washed three times with PFS and incubated with secondary antibodies diluted in PFS for 1 hr at RT, followed by washing twice in PFS and twice in PBS. Labelling was performed using Phalloidin (1:200) (Invitrogen) and Hoechst to label nuclei (10 µg/mL).

Acquisition of confocal images was performed using Opera Phenix Z9501 high-content imaging system (PerkinElmer), imaging at least 10 optical sections every 2 µM, imaging 25 fields at 20×. Images were analysed using Harmony imaging analysis software (PerkinElmer).

## Internalisation assay

Internalisation assays were performed as described in *Roberts et al., 2001*. Briefly, cells were surface labelled at 4℃ with 0.13 mg/mL NHS-SS-biotin (Pierce) in PBS for 30 min. Following surface labelling, cells were transferred to complete medium at 37℃ to allow internalisation in the presence of 0.6 mM primaquine for the indicated times. Biotin was then removed from the cell surface by treatment with the cell-impermeable-reducing agent MesNa. Cells were then lysed and the quantity of biotinylated receptors determined using a capture-ELISA. The following antibodies were used for capture-ELISA: clone VC5 (BDPharmingen, Cat# 555651) for $\alpha5\beta1$, anti-CD71 (BDPharmingen, Cat# 555534) for the TfnR, anti-HGFR (R&D Systems, Cat# AF276), and anti-EGFR1 (BDPharmingen, Cat# 555996).

## Statistical analysis

GraphPad Prism 8 software was used for statistical analyses. Information on sample size and statistical tests used for each experiment is indicated in the figure legends.

## Acknowledgements

We would like to thank Björn Kruspig, Sergi Marco, Martha Maria Zarou, Gaiti Hasan, Valerie Weaver, and Benoit Biteau for reagents and cell lines, Ann Hedley (CRUK Beatson) for help with bioinformatic analysis of the RNAseq data, and William Clark and Jillian Murray (CRUK Beartson) for cell line authentication and mycoplasma testing, respectively. We thank the Vienna Drosophila RNAi Center, the Bloomington Drosophila Stock Center, and the Developmental Studies Hybridoma Bank for providing *Drosophila* lines and reagents. MN was supported by a Leadership Fellowship from the University of Glasgow to (JBC). YY was supported by CRUK core funding to the CRUK Beatson Institute (A17196). The work from the Norman laboratory was funded by CRUK core funding for his laboratory (A18277), and JCN acknowledges the CRUK Glasgow Centre (C596/A18076). JJ, ADC, and OJS are funded by CRUK core funding for OJS laboratory (A21139). DMB, ARF, and ES are supported by the University of Glasgow and CRUK core funding (A17196). JBC is a Sir Henry Dale Fellow jointly funded by the Wellcome Trust and the Royal Society (grant number 104103/Z/14/Z).

# Additional information

## Competing interests

Owen J Sansom: O.J.S. has received funding from Novartis to examine RAL and RAL GEFs in malignancy. The other authors declare that no competing interests exist.

## Funding

| Funder | Grant reference number | Author |
|---|---|---|
| Wellcome Trust | 104103/Z/14/Z | Julia B Cordero |
| Cancer Research UK | A17196 | Yachuan Yu<br>Alvaro Román-Fernández<br>Emma Sandilands<br>David M Bryant |
| Cancer Research UK | A18277 | Jim C Norman |
| Cancer Research UK | C596/A18076 | Jim C Norman |

| Cancer Research UK | A21139 | Joel Johansson<br>Andrew D Campbell<br>Owen J Sansom |
| University of Glasgow | Leadership Fellowship | Máté Nászai |
| Royal Society | 104103/Z/14/Z | Julia B Cordero |

The funders had no role in study design, data collection and interpretation, or the decision to submit the work for publication.

## Author contributions

Máté Nászai, Data curation, Formal analysis, Investigation, Methodology, Writing - original draft, Writing - review and editing, Designed, performed and analysed most experiments; Karen Bellec, Data curation, Formal analysis, Investigation, Methodology, Writing - review and editing, Designed perform and analysed experiments required for the revision of the manuscript; Yachuan Yu, Emma Sandilands, Joel Johansson, Andrew D Campbell, Investigation, Methodology; Alvaro Román-Fernández, Data curation, Formal analysis, Investigation, Methodology; Jim C Norman, Formal analysis, Investigation, Methodology, Writing - review and editing, Designed, performed and analysed the EGFR internalisation experiments; Owen J Sansom, Formal analysis, Supervision, Writing - review and editing, Supervised the mouse intestinal regeneration experiment; David M Bryant, Data curation, Formal analysis, Supervision, Supervised 3D mammary tumour cell growth assays; Julia B Cordero, Conceptualization, Resources, Data curation, Formal analysis, Supervision, Funding acquisition, Writing - original draft, Project administration, Writing - review and editing

## Author ORCIDs

Owen J Sansom http://orcid.org/0000-0001-9540-3010
David M Bryant http://orcid.org/0000-0003-2721-5012
Julia B Cordero https://orcid.org/0000-0003-1701-9480

## Ethics

Animal experimentation: All animal work has been approved by a University of Glasgow internal ethics committee and performed in accordance with institutional guidelines under personal and project licenses granted by the UK Home Office (PPL PCD3046BA).

## Decision letter and Author response

Decision letter https://doi.org/10.7554/eLife.63807.sa1
Author response https://doi.org/10.7554/eLife.63807.sa2

# Additional files

## Supplementary files

• Supplementary file 1. Full genotype list. Table containing a list of all *Drosophila* genotypes used in the paper.

• Transparent reporting form

## Data availability

All data underlying the findings of this study are included in the manuscript and supporting file. Source data files have been provided for all figures containing numeric data. The entire raw dataset corresponding to the work in this paper is publicly available from our institutional repository at http://dx.doi.org/10.5525/gla.researchdata.1142. RNA sequencing data has been deposited in GEO (accession GSE162421) and can be accessed through https://www.ncbi.nlm.nih.gov/geo/query/acc.cgi?acc=GSE162421. Custom scripts used for quantification are available at Github: https://github.com/emltwc/TracheaProject/blob/master/Blind_scoring.ijm (copy archived at https://archive.softwareheritage.org/swh:1:rev:2ef7574e3c9bbb7ef852655511a86ef7531d35bb); https://github.com/emltwc/2018-Cell-Stem-Cell (copy archived at https://archive.softwareheritage.org/swh:1:rev:

e45f961ed6217ecc0bece566a76a633fd2b47ec0) and https://github.com/emltwc/EGFRProject (copy archived at https://archive.softwareheritage.org/swh:1:rev: 4888f27a6766694b33a8b25bcb42a078fa786f8d).

The following datasets were generated:

| Author(s) | Year | Dataset title | Dataset URL | Database and Identifier |
|---|---|---|---|---|
| Naszai M, Cordero JB | 2021 | RAL GTPases mediate EGFR/ MAPK signalling-driven intestinal stem cell proliferation and tumourigenesis upstream of RAS activation. | https://www.ncbi.nlm. nih.gov/geo/query/acc. cgi?acc=GSE162421 | NCBI Gene Expression Omnibus, GSE162421 |
| Nászai M, Bellec, K, Yu Y, Román-Fernández Á, Sandilands E, Johansson, J, Campbell A, Norman J, Sansom O, Bryant D, Cordero J | 2021 | RAL GTPases mediate EGFR-driven intestinal stem cell hyperproliferation and tumourigenesis | http://dx.doi.org/10. 5525/gla.researchdata. 1142 | Research Data, 10.5525/ gla.researchdata.1142 |

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

# Appendix 1

**Appendix 1—key resources table**

| Reagent type (species) or resource | Designation | Source or reference | Identifiers | Additional information |
|---|---|---|---|---|
| Strain, strain background (*Mus musculus*) | *VillinCreER* | *el Marjou et al., 2004* 10.1002/gene.20042 | NA | |
| Strain, strain background (*Mus musculus*) | *Rala^fl/fl* | *Peschard et al., 2012* 10.1016/j.cub.2012.09.013 | RRID:MGI:5505291 | |
| Strain, strain background (*Mus musculus*) | *Ralb^fl/fl* | *Peschard et al., 2012* 10.1016/j.cub.2012.09.013 | RRID:MGI:5505291 | |
| Strain, strain background (*Erwinia carotovora carotovora 15*) | *Ecc15* | B. Lemaitre; (*Basset et al., 2000*) 10.1073/pnas.97.7.3376 | NA | |
| Genetic reagent (*Drosophila melanogaster*) | *en>* | BDSC | RRID:BDSC_30564 | *y1 w*; P{w+mW.hs=en2.4 GAL4}e16E* |
| Genetic reagent (*Drosophila melanogaster*) | *ISC/EB>* | S. Hayashi; *Goto and Hayashi, 1999* PMID:10393119 | NA | *yw;esg-Gal4NP5130,UAS-GFP,UAS-GFPnLacZ/Cyo;tub-Gal80ts/Tm6B* |
| Genetic reagent (*Drosophila melanogaster*) | *Control* | R. Cagan | NA | *w[1118]* |
| Genetic reagent (*Drosophila melanogaster*) | *RalA-RNAi(1)* | VDRC | RRID:FlyBase_FBst0477124 | *P{KK108989}VIE-260B* |
| Genetic reagent (*Drosophila melanogaster*) | *RalA-RNAi(2)* | BDSC | RRID:BDSC_29580 | *y1 v1; P{y+t7.7v+t1.8=TRiP.JF03259}attP2* |
| Genetic reagent (*Drosophila melanogaster*) | *wg-RNAi* | VDRC | RRID:FlyBase_FBst0476437 | *P{KK108857}VIE-260B* |
| Genetic reagent (*Drosophila melanogaster*) | *wg-RNAi* | VDRC | RRID:FlyBase_FBst0450965 | *P{GD5007}v13351* |
| Genetic reagent (*Drosophila melanogaster*) | *EGFR-RNAi* | VDRC | RRID:FlyBase_FBst0478953 | *P{KK100051}VIE-260B* |

*Continued on next page*

*Appendix 1—key resources table continued*

| Reagent type (species) or resource | Designation | Source or reference | Identifiers | Additional information |
|---|---|---|---|---|
| Genetic reagent (*Drosophila melanogaster*) | *RalA*$^{wt}$ | G. Hasan; (**Richhariya et al., 2017**); 10.1038/srep42586 | NA | *P{UAS-RalA}3* |
| Genetic reagent (*Drosophila melanogaster*) | *GEFmeso-RNAi* | BDSC | RRID:BDSC_42545 | *y1 v1; P{y[+t7.7] v[+t1.8]=TRiP.HMJ02116}attP40* |
| Genetic reagent (*Drosophila melanogaster*) | *RalGPS-RNAi* | VDRC | RRID:FlyBase_FBst0463650 | *w[1118]; P{GD11683}v40596/TM3* |
| Genetic reagent (*Drosophila melanogaster*) | *Rgl-RNAi* | BDSC | RRID:BDSC_28938 | *y1 v1; P{y[+t7.7] v[+t1.8]=TRiP.HM05149}attP2* |
| Genetic reagent (*Drosophila melanogaster*) | *EGFR*$^{wt}$ | BDSC | RRID:BDSC_5368 | *y1 w[*]; P{w[+mc]=UAS Egfr.B}32-26-1* |
| Genetic reagent (*Drosophila melanogaster*) | *EGFR*$^{A887T}$ | BDSC | RRID:BDSC_9533 | *w[*]; P{w[+mC]=Egfr0.2.A887T.UAS}8-2* |
| Genetic reagent (*Drosophila melanogaster*) | *EGFR*$^{\lambda top}$ | BDSC | RRID:BDSC_59843 | *w[*]; P{w[+mC]=UAS Egfr.lambdatop}3/TM6C, Sb1* |
| Genetic reagent (*Drosophila melanogaster*) | *Ras*$^{V12}$*(1)* | BDSC | RRID:BDSC_64196 | *w[*]; P{w[+mC]=UAS-Ras85D.V12}2* |
| Genetic reagent (*Drosophila melanogaster*) | *Ras*$^{V12}$*(2)* | BDSC | RRID:BDSC_64195 | *w[*]; P{w[+mC]=UAS-Ras85D.V12}TL1* |
| Genetic reagent (*Drosophila melanogaster*) | *Ras-RNAi* | VDRC | RRID:FlyBase_FBst0478466 | *P{KK108029}VIE-260B* |
| Genetic reagent (*Drosophila melanogaster*) | *Src64*$^{wt}$ | BDSC | RRID:BDSC_8477 | *w[*]; P{w[+mC]=UAS-Src64B.C}2* |
| Cell line (*Homo sapiens*) | H1299 | ATCC CRL-5803 | RRID:CVCL_0060 | Authenticated through STR profiling *Mycoplasma* negative |
| Cell line (*Homo sapiens*) | HMT3522 T4-2 | V. Weaver, UCSF | RRID:CVCL_2501 | Authenticated through STR profiling *Mycoplasma* negative |
| Cell line (*Homo sapiens*) | HEK293-FT | Thermo Fisher Scientific | RRID:CVCL_6911 | Authenticated through STR profiling *Mycoplasma* negative |
| Antibody | Anti-GFP (Chicken polyclonal) | Abcam | RRID:AB_300798 | *Drosophila* IF (1:2000) |

*Continued on next page*

*Appendix 1—key resources table continued*

| Reagent type (species) or resource | Designation | Source or reference | Identifiers | Additional information |
|---|---|---|---|---|
| Antibody | Anti-Sox21a (Rabbit polyclonal) | B. Biteau; (**Meng and Biteau, 2015**) 10.1016/j.celrep.2015.09.061 | NA | *Drosophila* IF (1:2000) |
| Antibody | Anti-pERK (Rabbit polyclonal) | Cell Signalling Technology | RRID:AB_331646 | *Drosophila* IF (1:100); mouse IHC (1:400); western blot (1:1000) |
| Antibody | Anti-ERK (Rabbit polyclonal) | Cell Signalling Technology | RRID:AB_390779 | *Drosophila* IF (1:100); western blot (1:1000) |
| Antibody | Anti-ERK (Rabbit polyclonal) | Cell Signalling Technology | RRID:AB_330744 | Mouse IHC (1:40) |
| Antibody | Anti-rabbit IgG HRP-linked antibody (Goat polyclonal) | Cell Signalling Technology | RRID:AB_2099233 | Western blot (1:10,000) |
| Antibody | Anti-Phospho-Histone 3 Ser 10 (Rabbit polyclonal) | Cell Signalling Technology | RRID:AB_331535 | *Drosophila* IF (1:100) |
| Antibody | Anti-EGFR extracellular domain (Mouse monoclonal) | Sigma-Aldrich | RRID:AB_609900 | *Drosophila* IF (1:50) |
| Antibody | Anti-EGFR1 (Mouse monoclonal) | BDPharmingen | RRID:AB_2096589 | Capture-ELISA (5 µg/mL) |
| Antibody | Anti-c-MET (Goat polyclonal) | R&D Systems | RRID:AB_355289 | Capture-ELISA anti-HGFR (5 µg/mL) |
| Antibody | Anti-Alpha5 beta1 integrin (Mouse monoclonal, Clone V5) | BDPharmingen | RRID:AB_396007 | Capture-ELISA Anti-CD49e (5 µg/mL) |
| Antibody | Anti-Transferrin receptor (Human monoclonal) | BDPharmingen | RRID:AB_395918 | Capture-ELISA CD71 antibody (5 µg/mL) |
| Antibody | Alexa Fluor 488 anti-chicken-IgY (H + L) (Goat polyclonal secondary antibody) | Invitrogen | Cat#A-11039 RRID:AB_142924 | *Drosophila* IF (1:100) |
| Antibody | Alexa Fluor 594 anti-rabbit-IgG (H + L) (Goat polyclonal secondary antibody) | Invitrogen | Cat#A-11037 RRID:AB_2534095 | *Drosophila* IF (1:100) |
| Antibody | Alexa Fluor 594 anti-mouse-IgG (H + L) (Goat polyclonal secondary antibody) | Molecular Probes | RRID:AB_141672 | *Drosophila* IF (1:100) |

*Continued on next page*

*Appendix 1—key resources table continued*

| Reagent type (species) or resource | Designation | Source or reference | Identifiers | Additional information |
|---|---|---|---|---|
| Antibody | Alexa Fluor 594 anti-mouse-IgG (H + L) (Goat polyclonal secondary antibody) | Invitrogen | RRID:AB_2534091 | *Drosophila* IF (1:100) |
| Recombinant DNA reagent | pLKO.1-puromycin | Moffat et al. Cell. 2006 Mar 24. 124(6):1283–98 | RRID:Addgene_10878 | |
| Recombinant DNA reagent | VSVG | Trono lab, unpublished, donated to Addgene | RRID:Addgene_12259 | |
| Recombinant DNA reagent | SPAX2 | Trono lab, unpublished, donated to Addgene | RRID:Addgene_12260 | |
| Sequence-based reagent | Rho_Fwd | This paper | NA | TTGTCATCTTTGTCTCCTGCGA |
| Sequence-based reagent | Rho_Rev | This paper | NA | GTCAGGTGGGCAATGTACGA |
| Sequence-based reagent | Stg_Fwd | This paper | NA | CAGTAATAACACCAGCAGTTCGAG |
| Sequence-based reagent | Stg_Rev | This paper | NA | GTAGAACGACAGCTCCTCCT |
| Sequence-based reagent | Sox21a_Fwd | This paper | NA | AGACAATTAATACAGAGCTCGAGG |
| Sequence-based reagent | Sox21a_Rev | This paper | NA | GAGATGCTCGTCATGATGCC |
| Sequence-based reagent | Rpl32_Fwd | This paper | NA | AGGCCCAAGATCGTGAAGAA |
| Sequence-based reagent | Rpl32_Rev | This paper | NA | TGTGCACCAGGAACTTCTTGAA |
| Sequence-based reagent | Rala_Fwd | PrimerBank | ID#324072795 c2 | GCAGACAGCTATCGGAAGAAG |
| Sequence-based reagent | Rala_Rev | PrimerBank | ID#324072795 c2 | TCTCTAATTGCAGCGTAGTCCT |
| Sequence-based reagent | Ralb_Fwd | PrimerBank | ID#48762927 c1 | AGCCCTGACGCTTCAGTTC |
| Sequence-based reagent | Ralb_Rev | PrimerBank | ID#48762927 c1 | AGCGGTGTCCAGAATATCTATCT |

*Continued on next page*

*Appendix 1—key resources table continued*

| Reagent type (species) or resource | Designation | Source or reference | Identifiers | Additional information |
|---|---|---|---|---|
| Sequence-based reagent | *ActB_Fwd* | *Liu et al., 2015* 10.1371/journal.pone.0117058 | NA | TGACGTGGACATCCGCAAAG |
| Sequence-based reagent | *ActB_Rev* | *Liu et al., 2015* 10.1371/journal.pone.0117058 | NA | CTGGAAGGTGGACAGCGAGG |
| Sequence-based reagent | *shScr* | This paper | NA | CCGCAGGTATGCACGCGT |
| Sequence-based reagent | *shRala* | This paper | NA | GGAGGAAGTCCAGATCGATAT |
| Sequence-based reagent | *shRalb* | This paper | NA | CAAGGTGTTCTTTGACCTAAT |
| Sequence-based reagent | *siRNA Rala* (human) | Dharmacon | ONTARGETplus – Cat# L-009235-00-0005 | |
| Sequence-based reagent | *siRNA Ralb* (human) | Dharmacon | ONTARGETplus – Cat# L-008403-00-0005 | |
| Peptide, recombinant protein | EGF | Sigma | Cat# 11376454001 | |
| Peptide, recombinant protein | HGF | Sigma | Cat# H9661 | |
| Commercial assay or kit | High Capacity cDNA Reverse Transcription Kit | Applied Biosystems | Cat# 4368813 | |
| Commercial assay or kit | PerfeCTa SYBR Green FastMix (Low ROX) | Quanta Bio | Cat# 95074–012 | |
| Commercial assay or kit | VECTASHIELD Mounting Medium with DAPI | Vector Laboratories, Inc | RRID:AB_2336790 | |
| Commercial assay or kit | SuperSignal West Pico Chemiluminescent Substrate | Thermo Fisher Scientific | Cat# 34077 | |
| Commercial assay or kit | RNAeasy Mini Kit (50) | QIAGEN | Cat# 74104 | |
| Commercial assay or kit | Growth Factor Reduced Matrigel | BD Biosciences | 354230 | |
| Commercial assay or kit | Lipofectamine 2000 | Thermo Fisher Scientific | Cat# 11668027 | |
| Commercial assay or kit | Lenti-X Concentrator | Clontech | | |
| Chemical compound, drug | Glutamine | Thermo Fisher Scientific | 25030081 | |

*Continued on next page*

*Appendix 1—key resources table continued*

| Reagent type (species) or resource | Designation | Source or reference | Identifiers | Additional information |
|---|---|---|---|---|
| Chemical compound, drug | DMEM | Thermo Fisher Scientific | 12491015 | |
| Chemical compound, drug | FBS | Thermo Fisher Scientific | 26140079 | |
| Chemical compound, drug | L-Glutamine | Thermo Fisher Scientific | 25030081 | |
| Chemical compound, drug | Non-essential amino acids | Thermo Fisher Scientific | 11140050 | |
| Chemical compound, drug | Insulin | Sigma-Aldrich | I0516 | Insulin solution from bovine pancreas, 10 mg/mL insulin in 25 mm HEPES, pH 8.2, BioReagent, sterile-filtered, suitable for cell culture |
| Chemical compound, drug | Transferrin | Sigma-Aldrich | T2252 | |
| Chemical compound, drug | Sodium selenite | Sigma-Aldrich | S5261 | |
| Chemical compound, drug | β-Estradiol | Sigma-Aldrich | E2758 | |
| Chemical compound, drug | Hydrocortisone | Sigma-Aldrich | H0888 | |
| Chemical compound, drug | Prolactin | Miltenyi Biotech | 130-093-985 | |
| Chemical compound, drug | Tyrphostin-AG1478 | Sigma-Aldrich | T4182 | |
| Chemical compound, drug | Erlotinib, HCL | Sigma-Aldrich | SML2156 | |
| Chemical compound, drug | Puromycin | Thermo Fisher Scientific | A1113803 | |
| Chemical compound, drug | Phalloidin | Invitrogen | A12380, A22287 | |
| Chemical compound, drug | Hoechst | | H21486 | |
| Chemical compound, drug | RIPA buffer | Sigma | R0278 | |
| Chemical compound, drug | Bradford reagent | Abcam | AB119216 | |

*Continued on next page*

*Appendix 1—key resources table continued*

| Reagent type (species) or resource | Designation | Source or reference | Identifiers | Additional information |
|---|---|---|---|---|
| Chemical compound, drug | NuPAGE 10% Bis-Tris gel | Thermo Fisher Scientific | NP0301BOX | |
| Chemical compound, drug | NuPAGE MOPS SDS running buffer | | | |
| Chemical compound, drug | Trans-Blot Turbo PVDF membrane | Bio-Rad | 1704157 | |
| Chemical compound, drug | BSA | Sigma | A3294 | |
| Chemical compound, drug | Super Signal West Pico Chemiluminescent Substrate | Thermo Fisher Scientific | 34077 | |
| Software, algorithm | Fiji | NIH | | 1.51n; https://fiji.sc/ |
| Software, algorithm | GraphPad Prism 6 | GraphPad | RRID:SCR_002798 | |
| Software, algorithm | ZEN 2 lite | ZEISS | RRID:SCR_013672 | |
| Software, algorithm | 7500 Real-Time PCR Software | Applied Biosystems | RRID:SCR_014596 | |
| Software, algorithm | Harmony | PerkinElmer | | |
| Software, algorithm | BatchQuantify | (*Johansson et al., 2019*) 10.1016/j.stem.2019.02.002 | NA | https://github.com/emltwc/2018-Cell-Stem-Cell |
| Software, algorithm | EGFR_quant | This paper | NA | https://github.com/emltwc/EGFRProject |
| Software, algorithm | Blind scoring | (*Perochon et al., 2021*) https://doi.org/10.1038/s41556-021-00676-z | NA | https://github.com/emltwc/TracheaProject/blob/master/Blind_scoring.ijm |
| Other | Axio Observer | ZEISS | | |
| Other | LSM780 microscope | ZEISS | | |
| Other | BX51 microscope | Olympus | | |
| Other | Opera Phenix Z9501 | PerkinElmer | | |
| Other | 7500 Fast Real-Time PCR System | Applied Biosystems | | |
| Other | Trans-Blot Turbo system | Bio-Rad | 1704150 | |
| Other | HiSeq 2000 | Illumina | | |
| Other | ImageLock plate | Essen Biosciences | | |

