## [Decision Letter]

**Acceptance summary:**

By studying the small GTPase, RalA, in EGFR/MAPK signaling using both flies and human cancer cell lines you show that RalA determines the gain of EGFR signaling upstream of RAS activation. You document a role for RalA in internalization of ligand-bound EGFR, which is known to potentiate signaling downstream of the receptor. This is a new role for Ras related small GTPases in EGFR-MAPK signaling.

**Decision letter after peer review:**

Thank you for submitting your article "RAL GTPases mediate EGFR/MAPK driven intestinal stem cell proliferation and tumourigenesis upstream of RAS activation" for consideration by *eLife*. Your article has been reviewed by 3 peer reviewers, and the evaluation has been overseen by a Reviewing Editor and Utpal Banerjee as the Senior Editor. The following individuals involved in review of your submission have agreed to reveal their identity: Kartik Venkatachalam (Reviewer #1); Barry Thompson (Reviewer #2).

We would like to draw your attention to changes in our policy on revisions we have made in response to COVID-19 (https://elifesciences.org/articles/57162). Specifically, when editors judge that a submitted work as a whole belongs in *eLife* but that some conclusions require a modest amount of additional new data, as they do with your paper, we are asking that the manuscript be revised to either limit claims to those supported by data in hand, or to explicitly state that the relevant conclusions require additional supporting data.

We also enclose all the reviewers comments so you can address their concerns and issues. Please write a detailed rebuttal when you resubmit the manuscript.

*Reviewer #1:*

In this manuscript, the authors examine the role of the small GTPase, RalA, in EGFR/MAPK signaling using both flies and human cancer cell lines. They find that RalA determines the gain of EGFR signaling upstream of RAS activation. The authors then. attributed the regulation of EGFR/MAPK signaling to a role for RalA in internalization of ligand-bound EGFR, which is known to potentiate signaling downstream of the receptor. Strengths of the manuscript include an exciting and new description of Ras related small GTPases in determining EGFR-MAPK signaling and the utilization of a powerful combination of genetic interactions in flies and studies in human cancer cells. Indeed, the relevance of their finding to EGFR-driven malignancies is also a strength. Weaknesses are mostly technical and include the absence of key controls and need for improved assays for examining EGFR localization at the plasma membrane.

1. The data in Figure 1A would be more compelling if the authors quantify the reported changes. That would allow the reader to see the spread of the data. Also, they should overexpress wg and/or EGFR in the RalA knockdown wings to examine suppression of wing dysmorphia. This is needed to ascertain that the phenotypes in the RalA knockdown wings are related to those seen upon combined knockdown of wg and EGFR.

2. For the data shown in Figure 1C, it would be worthwhile to have positive controls showing that inhibition of EGFR/MAPK pathways in Ecc15-treated guts is sufficient to mitigate the upregulation of rho, stg, and Sox21a. The authors could do this genetically, or by using well established pharmacological inhibitors of MEK and ERK.

3. Does this change in pERK (as shown in Figure 1K) represent an increase in ERK phosphorylation, as would be expected upon the engagement of EGFR signaling? This question is also relevant to other figures in the manuscript that show pERK staining in the gut cells. The authors should examine the levels of total ERK as well to examine whether the elevation of pERK represents an increase in ERK phosphorylation. Alternatively (or in addition), the authors could check whether pharmacological inhibition of MEK decreases pERK levels. The most compelling data, of course, would be Western blots of fly tissues (perhaps the fat bodies or something similar) that examine both pERK and ERK upon expressing EGFR and/or RalA RNAi.

4. These data shown in Figures 4A-4D are not compelling. It is hard to appreciate PM localization of EGFR upon RalA knockdown. The authors most certainly need a significantly improved readout of EGFR PM localization in the fly cells. This concern also applies to data shown in other figures. Possibilities include staining of an extracellular epitope of EGFR in non-permeabilized cells and/or biotin labeling (similar to what they have done with H1299 cells.) In the event that these experiments are not possible in fly cells, the authors need to clearly explain why that is the case and use genetic strategies (e.g., dominant negative dynamin or something to that effect) to change the rates of receptor internalization and examine genetic interaction of these manipulations with RalA knockdown and overexpression.

5. Do the data shown in Figure 5 imply the purported effect of RalA on receptor internalization is specific for EGFR? Can the authors examine the localization of other receptors that signal to RAS/MAPK and see if this is dependent on RalA?

*Reviewer #2:*

This is an interesting manuscript that addresses a fundamental problem of the molecular regulation of intestinal stem cells in regeneration and in cancer, focusing on the role of the Ras-like (RAL) GTPases and their ability to regulate EGFR/MAPK signalling in the intestine. A novel molecular mechanism – regulation of EGFR internalisation – is identified. The experiments are well performed and the conclusions are justified by the data.

The major strengths include the combination of two model organisms, *Drosophila* and mammalian cells, which together provide compelling evidence for their novel molecular mechanism of RAL GTPase action in regulation of EGFR internalisation and signalling to p-ERK. Most of the genetic and mechanistic experiments are performed in *Drosophila*.

However, the role of RAL GTPases in regulating EGFR internalisation is also shown in mammalian cells by a surface biotinylation assay.

Given the importance of EGFR signalling in intestinal cancer, the results will be of broad interest to the community. In addition, RAL GTPases are potential therapeutic targets for cancer therapy.

*Reviewer #3:*

RAL GTPases are key effectors of EGFR signaling, a signaling pathway important for a variety of biological functions whose activity is highly regulated at multiple levels. The authors previously discovered a novel role for RAL in Wnt signaling pathway regulation through receptor trafficking. Here, they set out to investigate whether RAL also has a similar non-canonical role in regulating EGRF signaling.

Their main experimental system is the adult fly intestine where both Wnt and EGFR pathways have important and interdependent functions in regulating stem cell behavior. A key strength of the paper is that they validate their key findings in multiple (mammalian) experimental systems where EGFR signaling plays a key role and use a variety of experimental approaches. This rigorous approach increases both the significance and biological relevance of their findings.

Through their work, they demonstrate that:

1. RAL is necessary for the activation of EGFR/MAPK signaling following intestinal damage

2. Ectopic RAL is sufficient to activate EGFR/MAPK signaling

3. RAL is necessary for intestinal regeneration through increased stem cell division following intestinal damage

4. RAL regulates EGFR signaling by mediating EGF ligand induced EGFR endocytosis. Reduced RAL activity leads to reduced endocytosis, increased plasma membrane localization of EGFR and reduced signaling output.

All of these conclusions are well supported by their data. One key weakness of the manuscript is the claim that this is a non-canonical function of RAL that is upstream of RAS. The reader should consider alternative explanations for the results the authors base this claim on. The RasV12 phenotype shown in the manuscript is much stronger than that of ectopic EGFR and the RAL knockdown is unlikely to be complete. It is reasonable to conclude that RAL works still downstream of RAS to regulate EGFR localization as part of a feedback loop and that RasV12 phenotype is simply too strong to be rescued by a partial reduction of RAL activity. On a related note, placing RAL upstream of RAS in this particular context implies that RAS has a role downstream of RAL in EGFR internalization, which has not been demonstrated.

1. Additional evidence will be necessary to support the claim that EGFR internalization is a non-canonical role of RAL that is upstream of RAS. First, the authors should determine whether this function of RAL is independent of or downstream of RAS. For instance, does knocking down RAS has any effect on EGFR internalization? Testing additional RAS effectors downstream of RAL will also be helpful to determine whether EGRF internalization is regulated by the canonical RAL pathway downstream of RAS in a feedback loop.

The authors could also repeat the RAL knock down experiments using phenotypes induced by lower levels of ectopic RasV12 expression to obtain a phenotype comparable to that or ectopic EGFR for their genetic interaction experiments. They can increase the level of RAL knock down by introducing two copies of their RNA lines or combining the two independent RAL RNAi lines. These experiments will not provide a definitive answer so they not ideal but they can help support their claim if it is correct.

Alternatively, the authors could remove the claim that RAL is upstream of RAS from the paper (and the title) This would be acceptable to me as a reviewer since their findings are significant and relevant regardless of whether this claim is correct. In summary, this claim should either be removed or supported by additional data.

2. In Figure 1, sox21a appears to be also lost in non-GFP cells upon RAL knockdown. Possible explanations for this non-autonomous effect should be provided. A related point is that quantifications for this figure have been done within the ISC/EB compartment (GFP positive) for pERK staining and within the nuclear compartment of sox21a. It was not clear to me from looking at the figure annotations whether the authors looked at the nuclear compartment of all cells or GFP positive cells only This should be clarified.

3. In the introduction, the authors pose the question whether RAL proteins impact intestinal biology beyond Wnt signaling. As their analysis is largely focused on intestinal cell types with high Wnt signaling, it is understandably challenging to fully address this question in these cells. It would be interesting to include a discussion about the context dependence of their findings. Do they expect the role for RAL they uncovered to be unique to cells with relatively high WNT signaling? Do they expect to see a similar role for RAL in cells with no Wnt pathway activity? How does this relate to the potential of RAL as a drug target in different tumor types? What is known about the state of Wnt signaling in the mammalian cells they used? The authors can further contextualize their findings and increase the significance and impact of their work by elaborating on these questions in the discussion

4. Given the broad relevance of their work, the manuscript will also be of interest to scientists who are not deeply familiar with the details of RAS/RAL signaling. It would be helpful to include additional information regarding the canonical role of RAL downstream of RAS. Providing additional context would really emphasize the novel nature of their findings and help the work reach a broader audience.

---

## [Author Response]

Reviewer #1:In this manuscript, the authors examine the role of the small GTPase, RalA, in EGFR/MAPK signaling using both flies and human cancer cell lines. They find that RalA determines the gain of EGFR signaling upstream of RAS activation. The authors then. attributed the regulation of EGFR/MAPK signaling to a role for RalA in internalization of ligand-bound EGFR, which is known to potentiate signaling downstream of the receptor. Strengths of the manuscript include an exciting and new description of Ras related small GTPases in determining EGFR-MAPK signaling and the utilization of a powerful combination of genetic interactions in flies and studies in human cancer cells. Indeed, the relevance of their finding to EGFR-driven malignancies is also a strength. Weaknesses are mostly technical and include the absence of key controls and need for improved assays for examining EGFR localization at the plasma membrane.1. The data in Figure 1A would be more compelling if the authors quantify the reported changes. That would allow the reader to see the spread of the data. Also, they should overexpress wg and/or EGFR in the RalA knockdown wings to examine suppression of wing dysmorphia. This is needed to ascertain that the phenotypes in the RalA knockdown wings are related to those seen upon combined knockdown of wg and EGFR.

We now provide quantification of our wing phenotypes as requested by the Reviewer (Figure 1 A, B). Wing dysmorphia is a multi-dimensional phenotype, which includes defects in wing size and patterning. Therefore, we have quantified this phenotype using a blind scoring method previously developed in our lab (Perochon et al., 2021). Information of the methods used for quantification and associated macro is provided in the manuscript.

Furthermore, we have performed the additional genetic interactions suggested by the Reviewer. Unfortunately, overexpression of *wg* (*eng>wg*) led to early organismal lethality, which precluded us from performing the *eng>wg;RalA-RNAi* experiment. However, *EGFR* overexpression (*eng>EGFR*) was partially lethal and we were able to assess the role of RalA in this setting (Figure 3- supplement 1 A, B). Consistent with our genetic interactions in the adult midgut, *RalA* knockdown (*eng>EGFR; RalA-IR)* suppressed organismal lethality and adult wing vein patterning defects caused by EGFR overexpression (*eng>EGFR*) (Figure 3- supplement 1 A, B). We believe these results reinforce the critical role of RAL as an effector of EGFR signaling in multiple contexts.

2. For the data shown in Figure 1C, it would be worthwhile to have positive controls showing that inhibition of EGFR/MAPK pathways in Ecc15-treated guts is sufficient to mitigate the upregulation of rho, stg, and Sox21a. The authors could do this genetically, or by using well established pharmacological inhibitors of MEK and ERK.

Please note that *stg* and *Sox21a* have already been identified as transcriptional targets of *EGFR/MAPK* signaling within ISCs in the regenerating adult *Drosophila* midgut (Jin et al., 2015; Meng and Biteau, 2015). We have confirmed this through RT-qPCR of *esg^ts^>GFP* (control) and *esg^ts^>EGFR-RNAi* midguts mock treated or following *Ecc15* infection (Author response image 1). *Rho* is an important activator of EGFR signaling in the adult midgut. However, *rho* has been shown to work in cells other than those expressing the receptor, by potentiating ligand availability to the receptor (Liang et al., 2017; Ngo et al., 2020). While there’s is evidence in the embryo suggesting that *rho* expression can be indirectly regulated by EGFR/MAPK signaling (Hsu et al., 2001), we found no published evidence indicating *rho* as a target of EGFR/MAPK signaling in the midgut. It rather works upstream of EGFR activation. We have clarified this in the text. Consistently, knocking down EGFR does not impact upregulation of *rho* upon infection (Author response image 1). This indicates that the regulation of *rho* expression by RalA in the regenerating midgut is rather indirect/non-cell autonomous and it might represent a distinct mechanism of EGFR signaling regulation by RalA, which we don’t understand at the moment.

**Author response image 1. sa2fig1:** Transcriptional readouts of EGFR/MAPK pathway activation. RT-qPCR confirmation of transcriptional targets of EGFR/MAPK signalling in whole midguts from wild-type control and *EGFR* knockdown in stem/progenitor cells using *escargot-gal4, UAS-gfp.* Results are presented relative to rpl32 levels. n (number of biological replicates) = 4 or 3, each dot represents an independent RNA sample from >10 midguts per sample. Where indicated: *p<0.05, **p<0.01, ***p<0.001, ****p<0.0001; Two-way ANOVA followed by Sidak’s multiple comparisons test. All error bars represent SD.

3. Does this change in pERK (as shown in Figure 1K) represent an increase in ERK phosphorylation, as would be expected upon the engagement of EGFR signaling? This question is also relevant to other figures in the manuscript that show pERK staining in the gut cells. The authors should examine the levels of total ERK as well to examine whether the elevation of pERK represents an increase in ERK phosphorylation. Alternatively (or in addition), the authors could check whether pharmacological inhibition of MEK decreases pERK levels. The most compelling data, of course, would be Western blots of fly tissues (perhaps the fat bodies or something similar) that examine both pERK and ERK upon expressing EGFR and/or RalA RNAi.

This is indeed an important control, which we have now added to our studies. As we are not sure what is the relationship between RalA and EGFR in tissues like the fat body and to keep consistency with the rest of our studies, we have chosen to quantify total-ERK levels by immunofluorescence in the midgut in multiple contexts where we measured pERK levels. That is: in esg^ts^>w^1118^ Mock treated; esg^ts^>w^1118^ Ecc15 infected (Figure 1-supplement 1 I, J); esg^ts^>EGFR^wt^; esg^ts^>RalA-RNAi;EGFR^wt^ (Figure 3-supplement 1 E, F) and esg^ts^>RalA^wt^ fly midguts (Figure 1-supplement 1 F, G). Also, we now provide quantifications of pERK (Figure 1J) and total ERK (Figure 1-supplement 1E) in Wild type; Rala KO and Ralb KO damaged mouse intestines. We found no change in total ERK in either of these contexts, suggesting that our findings represent an effect of RAL GTPases in ERK activation.

Additionally, we have performed western blot of whole midguts from *esg>w^1118^ Mock treated; esg>w^1118^ Ecc15 infected* and *esg>RalA^wt^* fly midguts, which confirmed the results of the immunofluorescence (Figure 1-supplement 1H). We have only performed a subset of the genetic conditions for western blot as we have previously experienced that, while we can consistently and robustly detect the increase in basal levels of pERK in whole gut western blot (*e.g.* infected *vs* mock treated and control *vsRalA* overexpressing midguts) it is much more difficult to capture partial decrease in pERK levels as observed in situ (*e.g.* control infected *vsRalA-RNAi* infected midguts) from whole tissue preparation.

4. These data shown in Figures 4A-4D are not compelling. It is hard to appreciate PM localization of EGFR upon RalA knockdown. The authors most certainly need a significantly improved readout of EGFR PM localization in the fly cells. This concern also applies to data shown in other figures. Possibilities include staining of an extracellular epitope of EGFR in non-permeabilized cells and/or biotin labeling (similar to what they have done with H1299 cells.) In the event that these experiments are not possible in fly cells, the authors need to clearly explain why that is the case and use genetic strategies (e.g., dominant negative dynamin or something to that effect) to change the rates of receptor internalization and examine genetic interaction of these manipulations with RalA knockdown and overexpression.

We appreciate the Reviewer’s concerns regarding EGFR membrane localization in the fly gut. Unfortunately, there are significant technical challenges to the experimental approaches suggested by the Reviewer. The EGFR antibody we have used in our *Drosophila* immunostaining indeed recognises an extracellular epitope of the receptor. However, our attempts to do immunostaining of the fly gut epithelium without tissue permeabilization have not been successful. The problem is most likely due to the layers of visceral muscle and basement membrane surrounding the basal side of the gut tube, which prevents antibody from penetrating into the epithelium in the absence of permeabilization. This is precisely one of the reasons why we have complemented our fly work with the biochemical studies in cell lines, which quantitatively assess EGFR internalization. We also tried performing co-localization studies between EGFR and cell-junction proteins. However, the best available antibody to label cell junctions in ISCs/EBs is anti-Armadillo/β-Catenin, which is unfortunately the same species as the EGFR antibody, precluding their combined use. As an alternative, we have tried co-staining experiments using anti-EGFR and anti-E-cadherin or anti-Discs Large but none of the two latter showed robust labelling of ISC/EBs to produce better data than what we have.

As an additional attempt to address the reviewers concern, we have tried to perform the genetic interactions suggested. However, consistent with published reports (Nagy et al., 2016; Zhang et al., 2019), global blockade of endocytosis in ISCs/EBs by overexpression of dominant negative *Rab5* (*esg^ts^>Rab5^DN^*) or temperature sensitive dominant negative dynamin (*Shibire*) *(esg>Shi^ts^*) induces massive intestinal hyperplasia (Author response image 2), which precludes their use in meaningful genetic interactions. Admittedly, these genetic approaches are much broader than knocking down *RalA* and likely lead to pleotropic effects in signaling activity amongst other things.

**Author response image 2. sa2fig2:** Inhibition of cell membrane internalisation induces hyperproliferation. Representative confocal images of adult posterior midguts from wild-type control animals, or following the overexpression of temperature sensitive dominant negative dynamin (*shi^DN^*), or dominant negative Rab5 (*Rab5^DN^*) in stem/progenitor cells using *escargot-gal4, UAS-gfp* (*ISC/EB>*; green). Scale bar = 50 µ. Quantification of pH3 counts, as in A, in guts expressing dominant negative dynamin. Quantification of pH3 counts, as in A, in guts expressing dominant negative Rab5. Where indicated: *p<0.05, **p<0.01, ***p<0.001, ****p<0.0001; Student’s t-test. All error bars represent SD.

We have used CD8-GFP driven by *escargot-Gal4* (*esg>GFP*) to label cell membranes and estimate EGFR localization in our original studies. As a means to improve this part of our manuscript and address as best as we can the Reviewer’s concern, we now provide improved images and also a detailed account of the approach and criteria used to define membrane versus intracellular EGFR localization in our system (Figure 4; Figure 4-supplement 1 and 2; Figure 5G)

Specifically, we now provide:

a. Representative single z-stack slice of the original data.

b. Pseudo-coloured image using the turbo colour map to better highlight intensity differences.

c. Closeup of multiple regions of interest.

5. Do the data shown in Figure 5 imply the purported effect of RalA on receptor internalization is specific for EGFR? Can the authors examine the localization of other receptors that signal to RAS/MAPK and see if this is dependent on RalA?

As shown in our previous paper, RAL GTPases are involved in the internalisation of the Wnt receptor Frizzled (Johansson et al., 2019). Here, we show that RALs are also necessary for the internalization of EGFR. Our experiments in H1299 cells show that steady state turnover of α5β1 integrin, human transferrin receptor or indeed EGFR was not affected (Figure 4-supplement 3 E, F). However, ligand-induced EGFR internalisation was significantly reduced in the absence of RAL GTPases (Figure 4G). We now include additional receptor localisation data on the receptor tyrosine kinase c-Met that also signals through RAS/MAPK when exposed to its ligand, Hepatocyte Growth Factor (HGF). Unlike with EGFR, *Ral* knockdown had no impact on HGF-driven internalisation of cMet (Figure 4-supplement 3 C, D). This evidence points to at least some degree of specificity in the action of RAL proteins on signaling receptor internalization. We have now discussed this in the appropriate Results section.

Reviewer #3:RAL GTPases are key effectors of EGFR signaling, a signaling pathway important for a variety of biological functions whose activity is highly regulated at multiple levels. The authors previously discovered a novel role for RAL in Wnt signaling pathway regulation through receptor trafficking. Here, they set out to investigate whether RAL also has a similar non-canonical role in regulating EGRF signaling.Their main experimental system is the adult fly intestine where both Wnt and EGFR pathways have important and interdependent functions in regulating stem cell behavior. A key strength of the paper is that they validate their key findings in multiple (mammalian) experimental systems where EGFR signaling plays a key role and use a variety of experimental approaches. This rigorous approach increases both the significance and biological relevance of their findings.Through their work, they demonstrate that:1. RAL is necessary for the activation of EGFR/MAPK signaling following intestinal damage2. Ectopic RAL is sufficient to activate EGFR/MAPK signaling3. RAL is necessary for intestinal regeneration through increased stem cell division following intestinal damage4. RAL regulates EGFR signaling by mediating EGF ligand induced EGFR endocytosis. Reduced RAL activity leads to reduced endocytosis, increased plasma membrane localization of EGFR and reduced signaling output.All of these conclusions are well supported by their data. One key weakness of the manuscript is the claim that this is a non-canonical function of RAL that is upstream of RAS. The reader should consider alternative explanations for the results the authors base this claim on. The RasV12 phenotype shown in the manuscript is much stronger than that of ectopic EGFR and the RAL knockdown is unlikely to be complete. It is reasonable to conclude that RAL works still downstream of RAS to regulate EGFR localization as part of a feedback loop and that RasV12 phenotype is simply too strong to be rescued by a partial reduction of RAL activity. On a related note, placing RAL upstream of RAS in this particular context implies that RAS has a role downstream of RAL in EGFR internalization, which has not been demonstrated.

These are all very valid criticisms by the Reviewer. Our statement in the original title and text refers to the fact that ISC hyperproliferation and MAPK activation caused upon activation of EGFR but not Ras activation are sensitive to RAL presence. We have now modified the title and worked through the text to better reflect on this and avoid misinterpretation. The potential effect of differential phenotypic strength resulting from the various genetic manipulations to hyperactivate EGFR/MAPK signaling is one that we have worked hard at addressing in our manuscript:

a. We consistently used Dicer to improve RNAi efficiency in all the genetic combinations.

b. We have kept a consistent timing of overexpression for all the transgenes used.

c. Overexpression of constitutively active forms of EGFR leads to levels of ISC proliferation much higher than those upon wild-type EGFR overexpression and very much comparable or even higher than those observed upon *Ras^v12^* overexpression (Figure 3A, B and E, F; Figure 3-supplement 1 C, D). Nevertheless, only ISC hyperproliferation driven by constitutively active EGFR is sensitive to *RalA* knockdown (Figure 3A, B and E, F; Figure 3-supplement 1 C, D). Importantly, the *RalA-RNAi* used in these contexts was the same.

d. We also assessed the effect of *RalA* knockdown on the overexpression of two independent *Ras^v12^* constructs, which led to significantly different levels of ISC proliferation and did not observe an effect of *RalA* knockdown in either case (Figure 3A, B; Figure 3-supplement 1 C, D).

Altogether, we believe this evidence strongly supports a role of *RalA* mediating intestinal hyperplasia caused by hyperactivation of EGFR but not upon constitutively active Ras. However, we agree we need to avoid a general statement of non-canonical action of RAL and be more specific of what we refer to when we place RAL upstream of Ras activation.

1. Additional evidence will be necessary to support the claim that EGFR internalization is a non-canonical role of RAL that is upstream of RAS. First, the authors should determine whether this function of RAL is independent of or downstream of RAS. For instance, does knocking down RAS has any effect on EGFR internalization? Testing additional RAS effectors downstream of RAL will also be helpful to determine whether EGRF internalization is regulated by the canonical RAL pathway downstream of RAS in a feedback loop.

This very valid point. As suggested by the Reviewer, we have now tested and confirmed that, indeed, Ras also influences EGFR membrane localization in the midgut in (Figure 4-supplement 2). Our statement in the original title and abstract refers to the fact that ISC proliferation and MAPK activation caused upon *Ras* hyperactivation is refractory to RAL’s action. We have now modified the title and the manuscript, including removing our *‘non-canonical role of RAL’* statement to more accurately represent the conclusions of our results. We have also included a working model figure.

The authors could also repeat the RAL knock down experiments using phenotypes induced by lower levels of ectopic RasV12 expression to obtain a phenotype comparable to that or ectopic EGFR for their genetic interaction experiments. They can increase the level of RAL knock down by introducing two copies of their RNA lines or combining the two independent RAL RNAi lines. These experiments will not provide a definitive answer so they not ideal but they can help support their claim if it is correct.Alternatively, the authors could remove the claim that RAL is upstream of RAS from the paper (and the title) This would be acceptable to me as a reviewer since their findings are significant and relevant regardless of whether this claim is correct. In summary, this claim should either be removed or supported by additional data.

As detailed in points a-d of our first response to the Reviewer, we have taken several steps to ensure to the best of our capabilities that the differential impact of *RalA* knockdown in *EGFR* versus *Ras* hyperactivation is not a consequence of differential phenotypic strength. As pointed by the Reviewer, while not perfect, we believe the evidence presented in this regard helps support our claim that RalA mediates EGFR but not *Ras* driven ISC hyperproliferation. We have additionally modified the title and text to more clearly present our results.

2. In Figure 1, sox21a appears to be also lost in non-GFP cells upon RAL knockdown. Possible explanations for this non-autonomous effect should be provided. A related point is that quantifications for this figure have been done within the ISC/EB compartment (GFP positive) for pERK staining and within the nuclear compartment of sox21a. It was not clear to me from looking at the figure annotations whether the authors looked at the nuclear compartment of all cells or GFP positive cells only This should be clarified.

Thank you for this observation. The figure panel in question (Figure 1E; mock treated control) that was originally included in our submitted manuscript has some IF staining artefacts. Upon closer examination, what appears to be staining in non-GFP cells is background specs and non-specific labelling of tracheal tissue outside of the nuclear compartment. Our quantification of Sox21a measures staining intensity within the nuclei of all cells, therefore this noise is not included. The image was changed to a different picture from the same experiment that did not have the staining artefacts. Our methods have now been updated to clarify what compartments are quantified in each experiment.

3. In the introduction, the authors pose the question whether RAL proteins impact intestinal biology beyond Wnt signaling. As their analysis is largely focused on intestinal cell types with high Wnt signaling, it is understandably challenging to fully address this question in these cells. It would be interesting to include a discussion about the context dependence of their findings. Do they expect the role for RAL they uncovered to be unique to cells with relatively high WNT signaling? Do they expect to see a similar role for RAL in cells with no Wnt pathway activity? How does this relate to the potential of RAL as a drug target in different tumor types? What is known about the state of Wnt signaling in the mammalian cells they used? The authors can further contextualize their findings and increase the significance and impact of their work by elaborating on these questions in the discussion.

A report from the literature shows that H1299 are proficient to activate Wnt signaling upon addition of exogenous Wnt3a (Melnik et al., 2018). However, we have not added Wnt3a in our experiments. Importantly, we observed that the effect of RAL proteins on EGFR internalization in these cells requires the addition of EGF ligand (Figure 4G). Therefore, in this context, it seems that the role of RAL is linked to high levels of EGFR signaling.

Similarly, the growth of HMT-3522 T4-2 cells is heavily dependent on EGFR activation and they are highly sensitive to EGFR inhibitors (Madsen et al., 1992; Wang et al., 1998) (Figure 5I, J). Therefore, our results do suggest a potential role of RalA and value as a therapeutic target in contexts beyond the intestine and upon high levels of EGFR activity. In fact, we have previously shown that loss of *Apc* in the intestine, the most predominant mutation leading to intestinal malignancy through constitutive activation of Wnt signaling, is refractive to RAL inhibition (Johansson et al., 2019). We included some of this information in the discussion but have now expanded on these points as suggested by the Reviewer.

4. Given the broad relevance of their work, the manuscript will also be of interest to scientists who are not deeply familiar with the details of RAS/RAL signaling. It would be helpful to include additional information regarding the canonical role of RAL downstream of RAS. Providing additional context would really emphasize the novel nature of their findings and help the work reach a broader audience.

A report from the literature shows that H1299 are proficient to activate Wnt signaling upon addition of exogenous Wnt3a (Melnik et al., 2018). However, we have not added Wnt3a in our experiments. Importantly, we observed that the effect of RAL proteins on EGFR internalization in these cells requires the addition of EGF ligand (Figure 4G). Therefore, in this context, it seems that the role of RAL is linked to high levels of EGFR signaling.

Similarly, the growth of HMT-3522 T4-2 cells is heavily dependent on EGFR activation and they are highly sensitive to EGFR inhibitors (Madsen et al., 1992; Wang et al., 1998) (Figure 5I, J). Therefore, our results do suggest a potential role of RalA and value as a therapeutic target in contexts beyond the intestine and upon high levels of EGFR activity. In fact, we have previously shown that loss of *Apc* in the intestine, the most predominant mutation leading to intestinal malignancy through constitutive activation of Wnt signaling, is refractive to RAL inhibition (Johansson et al., 2019). We included some of this information in the discussion but have now expanded on these points as suggested by the Reviewer.

References:

Hsu, T., McRackan, D., Vincent, T. S., and Gert de Couet, H. (2001). *Drosophila* Pin1 prolyl isomerase Dodo is a MAP kinase signal responder during oogenesis. Nature Cell Biology, *3*(6), 538–543. https://doi.org/10.1038/35078508

Jin, Y., Ha, N., Forés, M., Xiang, J., Gläßer, C., Maldera, J., Jiménez, G., and Edgar, B. A. (2015). EGFR/Ras Signaling Controls *Drosophila* Intestinal Stem Cell Proliferation via Capicua-Regulated Genes. PLOS Genetics, *11*(12), e1005634. https://doi.org/10.1371/journal.pgen.1005634

Johansson, J., Naszai, M., Hodder, M. C., Pickering, K. A., Miller, B. W., Ridgway, R. A., Yu, Y., Peschard, P., Brachmann, S., Campbell, A. D., Cordero, J. B., and Sansom, O. J. (2019). RAL GTPases Drive Intestinal Stem Cell Function and Regeneration through Internalization of WNT Signalosomes. Cell Stem Cell, *24*(4), 592-607.e7. https://doi.org/10.1016/j.stem.2019.02.002

Liang, J., Balachandra, S., Ngo, S., and O’Brien, L. E. (2017). Feedback regulation of steady-state epithelial turnover and organ size. Nature, *548*(7669), 588–591. https://doi.org/10.1038/nature23678

Madsen, M. W., Lykkesfeldt, A. E., Laursen, I., Nielsen, K. V, and Briand, P. (1992). Altered gene expression of c-myc, epidermal growth factor receptor, transforming growth factor-α, and c-erb-B2 in an immortalized human breast epithelial cell line, HMT-3522, is associated with decreased growth factor requirements. Cancer Research, *52*(5), 1210–1217. http://www.ncbi.nlm.nih.gov/pubmed/1737382

Melnik, S., Dvornikov, D., Müller-Decker, K., Depner, S., Stannek, P., Meister, M., Warth, A., Thomas, M., Muley, T., Risch, A., Plass, C., Klingmüller, U., Niehrs, C., and Glinka, A. (2018). Cancer cell specific inhibition of Wnt/β-catenin signaling by forced intracellular acidification. *Cell Discovery*, *4*(1), 37. https://doi.org/10.1038/s41421-018-0033-2

Meng, F. W., and Biteau, B. (2015). A Sox Transcription Factor Is a Critical Regulator of Adult Stem Cell Proliferation in the *Drosophila* Intestine. Cell Reports, *13*(5), 906–914. https://doi.org/10.1016/j.celrep.2015.09.061

Nagy, P., Kovács, L., Sándor, G. O., and Juhász, G. (2016). Stem-cell-specific endocytic degradation defects lead to intestinal dysplasia in *Drosophila*. Disease Models and Mechanisms, *9*(5), 501–512. https://doi.org/10.1242/dmm.023416

Ngo, S., Liang, J., Su, Y.-H., and O’Brien, L. E. (2020). Disruption of EGF Feedback by Intestinal Tumors and Neighboring Cells in *Drosophila*. Current Biology, *30*(8), 1537-1546.e3. https://doi.org/10.1016/j.cub.2020.01.082

Perochon, J., Yu, Y., Aughey, G. N., Medina, A. B., Southall, T. D., and Cordero, J. B. (2021). Dynamic adult tracheal plasticity drives stem cell adaptation to changes in intestinal homeostasis in *Drosophila*. Nature Cell Biology, *23*(5), 485–496. https://doi.org/10.1038/s41556-021-00676-z

Wang, F., Weaver, V. M., Petersen, O. W., Larabell, C. A., Dedhar, S., Briand, P., Lupu, R., and Bissell, M. J. (1998). Reciprocal interactions between beta1-integrin and epidermal growth factor receptor in three-dimensional basement membrane breast cultures: a different perspective in epithelial biology. Proceedings of the National Academy of Sciences of the United States of America, 9*5*(25), 14821–14826. https://doi.org/10.1073/pnas.95.25.14821

Zhang, P., Holowatyj, A. N., Roy, T., Pronovost, S. M., Marchetti, M., Liu, H., Ulrich, C. M., and Edgar, B. A. (2019). An SH3PX1-Dependent Endocytosis-Autophagy Network Restrains Intestinal Stem Cell Proliferation by Counteracting EGFR-ERK Signaling. Developmental Cell, *49*(4), 574-589.e5. https://doi.org/10.1016/j.devcel.2019.03.029